# JOINTLY TRAINING TASK-SPECIFIC ENCODERS AND DOWNSTREAM MODELS ON HETEROGENEOUS MULTIPLEX GRAPHS

## ABSTRACT

Learning representations on Heterogeneous Multiplex Graphs (HMGs) is an active field of study, driven by the need for generating expressive, low-dimensional embeddings to support downstream machine learning tasks. A key component of this process is the design of the graph processing pipeline, which directly impacts the quality of learned representations. Information fusion techniques, which aggregate information across layers of a multiplex graph, have been shown to improve the performance of Graph Neural Network (GNN)-based architectures on various tasks including node classification, edge prediction, and graph-level classification. Recent research has explored fusion strategies at different stages of the processing pipeline, leading to graph-, GNN-, embedding-, and prediction-level approaches. In this work, we propose a model extending the `GraphSAGE` architecture, which simultaneously refines layer-wise embeddings produced by the encoder while training downstream models. We evaluate the model's effectiveness on an HMG on real-world and benchmark datasets, comparing it to models utilizing either graph-level or prediction-level fusion without jointly optimizing their vector embeddings. We demonstrate that our approach enhances the model's performance on downstream tasks, particularly node classification.

## 1 INTRODUCTION

As the internet continues to expand rapidly, users are increasingly exposed to irrelevant and unwanted information, driving demand for systems connecting users with more personalized and streamlined content (Bawden & Robinson (2020)). Recommendation systems have thus become essential for enhancing user experiences, and for enabling companies to influence consumer behavior across industries. These systems learn and predict user preferences to recommend relevant items, driving user engagement, satisfaction, and revenue (Roy & Dutta (2022)). In 2013, 35% of Amazon purchases and 75% of Netflix streams were driven by algorithmic recommendations (MacKenzie (2024)). Their significance is underscored by the growth of companies who rely on them including Amazon, Netflix, Spotify, and Facebook.

Recommendation systems are categorized as either content-based, collaborative filtering-based, or hybrid systems (Roy & Dutta (2022)). Content-based algorithms create user and item profiles from features and interactions, but face shortcomings with exploring new interests and leveraging the information of other users. Collaborative filtering addresses this problem by creating neighbourhoods of users with similar interests, using either memory- or model-based approaches to make predictions (Roy & Dutta (2022)).

These approaches typically operate only on predefined data types and user-item connections. With advances in data collection, modern recommendation systems aim to incorporate more diverse and complex feature and connection data. This demand has led to the popularization of hybrid- and graph- based systems, which excel at modeling intricate relationships. (Roy & Dutta (2022)).

Graph Representation Learning (GRL) algorithms use shallow or deep methods to learn low-dimensional embedding vectors of graph elements, enabling their use for downstream tasks including node classification, edge prediction, clustering, and graph-level classification (Kipf & Welling (2016); Hamilton (2020)). While most GNN-based architectures are designed for homogeneous

graphs, with uniform node and edge types, real-world networks are often inherently heterogeneous and/or multiplex, containing both nodes and edges of multiple distinct types (Zhang et al. (2024)). Accordingly, improving learned representations for Heterogeneous Multiplex Graphs (HMGs) is an active field of research.

## 1.1 MULTIPLEX GRAPH REPRESENTATION LEARNING

Multiplex graphs model different connection types between nodes as distinct edge layers, enabling the incorporation of edge type information into models (Gong & Cheng (2019)). Information fusion is necessary to consolidate information across layers of a multiplex network during representation learning, and is categorized into four distinct types: graph-, GNN-, embedding-, and prediction-level Bielak & Kajdanowicz (2024).

**Graph-level fusion**-based architectures flatten all edge layers of a multiplex graph into a single layer prior to learning embeddings. Models such as MHGCN (Yu et al. (2022)) learn aggregation functions, while others simply flatten the graph before passing it to a GNN, Graph Convolutional Network (GCN), or other homogeneous graph-compatible model. These architectures are typically outperformed by models utilizing more complex fusion strategies (Bielak & Kajdanowicz (2024)).

**GNN-level fusion**-based architectures consider the multi-layered graph structure while learning node embeddings, to account for connections in all layers. DMGI combines layer-wise embeddings using a learned matrix (Park et al. (2020)). HDGI uses an attention mechanism to capture layer-specific information (Ren et al. (2020)). $S^2$MGRL uses a Multi-Layer-Perceptron (MLP) and GCN to provide a fused embedding that captures layer-specific information (Mo et al. (2022)). These methods output a single embedding for each node to be used in downstream tasks.

**Embedding-level fusion**-based models compute node embeddings for each layer and then combine them using a fusion operator. Operators can be trainable (using attention, projection, or matrix lookup) or non-trainable (Mean, Concat, Min, Max, or Sum) (Bielak & Kajdanowicz (2024)). Complex architectures, such as MxPool, use trained, differentiable pooling operators to generate hierarchical node embeddings (Liang et al. (2020)).

**Prediction-level fusion** learns node embeddings independently for each layer using an architecture like DGI, DW, ARVGA, or `GraphSAGE`, and trains a distinct task model on each layer.(Veličković et al. (2018); Perozzi et al. (2014); Pan et al. (2018); Hamilton et al. (2017)). Predictions are aggregated across layers using either soft or hard ensemble voting to give a single result (Bielak & Kajdanowicz (2024)).

These information fusion techniques produce node embeddings for multiplex graphs, but are not specifically optimized for the downstream tasks for which they may be used.

## 1.2 USING LEARNED REPRESENTATIONS FOR DOWNSTREAM TASKS

Learned embeddings can perform sub-optimally on downstream tasks such as node classification and clustering (Veličković et al. (2018); Hamilton (2020)). To address these challenges, joint optimization, where encoders and downstream models are trained simultaneously have been used in other application domains such as computer vision and text classification (Zhuge et al. (2019); Jernite et al. (2017); Asano et al. (2020); Vargas-Vieyra et al. (2020)). This paradigm has been shown to produce task-optimal representations and improve downstream performance relative to baseline models.

The objective of this paper is three-fold:

1. To apply joint-optimization techniques for learning task-specific representations for graph-based recommendation systems.

2. Handle the challenges of learning meaningful representations in the context of HMGs.

3. Provide a set of best practices for learning representations from realistic graph datasets.

**Simultaneous Encoder-Classifier Optimization (SECO)** To address objective 1, we introduce and define the concept of SECO, a paradigm in which a model iteratively and simultaneously up-

| Model | Information Fusion Level | | | | SECO |
|---|---|---|---|---|---|
| | Graph- | GNN- | Embedding- | Prediction- | |
| Graph-level flattening (GCN/GAT/DW/DGI) (Bielak & Kajdanowicz (2024)) | ✓ | ✗ | ✗ | ✗ | ✗ |
| MHGCN (Yu et al. (2022)) | ✓ | ✗ | ✗ | ✗ | ✗ |
| Model C1 | ✓ | ✗ | ✗ | ✗ | ✗ |
| Model C2 | ✓ | ✗ | ✗ | ✗ | ✓ |
| DMGI (Park et al. (2020)) | ✗ | ✓ | ✗ | ✗ | ✗ |
| HDGI (Ren et al. (2020)) | ✗ | ✓ | ✗ | ✗ | ✗ |
| MxPool (Liang et al. (2020)) | ✗ | ✓ | ✗ | ✗ | ✗ |
| S^2MGRL (Mo et al. (2022)) | ✗ | ✓ | ✗ | ✗ | ✗ |
| Embedding-level flattening (GCN/GAT/DW/DGI) (Bielak & Kajdanowicz (2024)) | ✗ | ✗ | ✓ | ✗ | ✗ |
| Vote (Bielak & Kajdanowicz (2024)) | ✗ | ✗ | ✗ | ✓ | ✗ |
| Model C3 | ✗ | ✗ | ✗ | ✓ | ✗ |
| **SECSAGE** - (Our proposed model) | ✗ | ✗ | ✗ | ✓ | ✓ |

Table 1: Comparison of Information Fusion methods on Multiplex GRL Models

dates its encoder and downstream classifier in a supervised manner, learning task-optimal node embeddings.

**HMGSAGE**  To address the second objective of the paper, we introduce `HMGSAGE`, an encoder extending the `GraphSAGE` architecture for inductive GRL to learn embeddings on HMGs. The encoder uses parallel `GraphSAGE` layers to sample and aggregate neighbours of nodes within the graph, preserving the structure of the data while maintaining flexibility and scalability on large graphs. `HMGSAGE` can be trained as part of a Graph Autoencoder (GAE) using contrastive approaches to learn node embeddings for HMGs that are useful for edge prediction tasks.

**SECSAGE**  This model combines the `HMGSAGE` encoder with a downstream classifier and aggregates the classification outputs using prediction-level fusion. `HMGSAGE` and the classifier are trained simultaneously using SECO, jointly optimizing embeddings learned by the `HMGSAGE` encoder for node classification tasks.

Studying these concepts and models addresses the existing research gap in the study of SECO. We show that it increases a classifier's Macro F1 accuracy, and produces more expressive and separable embeddings, as evidenced by their silhouette scores. The performance of both `HMGSAGE` and `SECSAGE` are assessed and benchmarked against baseline models for edge prediction and node classification, respectively. First, `HMGSAGE` is tested against a `GraphSAGE` encoder operating on a flattened graph. Second, `SECSAGE` is evaluated against Models C1, C2, and C3 defined in Table 1 on node classification tasks. Both `HMGSAGE` and `SECSAGE` are shown to outperform baseline models, and SECO is shown to reduce training time compared to methods where encoders and classifiers are trained separately.

We address the third objective of our paper by testing the concepts defined above on a real-world travel dataset, "Travel Dubai", and a standard "Amazon" product review dataset (Hou et al. (2024)) while highlighting the challenges associated with these datasets such as underrepresentation of class labels and multiplex layers.

## 2 METHODOLOGY

We first define the following notation related to HMGs and the concept of inductive representation learning to help describe the proposed models in Section 1.2.

**Definition 1.** **Heterogeneous Graph** A heterogeneous graph is defined as a graph $\mathcal{G} = (\mathcal{V}, \mathcal{E}, \mathcal{Q}, \phi)$, in which $\mathcal{V}$ is a set of vertices, $\mathcal{E}$ is a set of edges, and $\phi$ is a vertex type mapping function $\phi : \mathcal{V} \to \mathcal{Q}$. In a heterogeneous graph, the set of vertices can be partitioned into k disjoint sets $\mathcal{V} = \mathcal{V}_1 \cup \mathcal{V}_2 \cup ... \cup \mathcal{V}_k$ where $k = |\mathcal{Q}|, \mathcal{V}_i \bigcap \mathcal{V}_j = \emptyset, \forall i \neq j$ and each set of vertices $\mathcal{V}_i$ represents a distinct type of node with a unique set of properties (Hamilton (2020)).

**Definition 2. Multiplex Graph** A multiplex graph is a graph $\mathcal{G} = (\mathcal{V}, \mathcal{E}_\tau, R, \psi), \forall \tau \in \mathcal{R}$, where $\psi : \mathcal{E} \to \mathcal{R}$ maps each edge to its type. The set of edges for each layer of the graph, $\mathcal{E}_\tau$, is a subset of the Cartesian product of all vertices in the graph, $\mathcal{E}_\tau \subseteq \mathcal{V} \times \mathcal{V}, \forall \tau \in \mathcal{R}$. Thus, there are multiple layers of connections, and for any pair of vertices $u, v \in \mathcal{V}$, up to one connection in each edge layer $\mathcal{E}_\tau$ is permitted.

**Definition 3. Heterogeneous Multiplex Graph** In this paper, a Heterogeneous Multiplex Graph (HMG) is defined as a graph $\mathcal{G} = (\mathcal{V}, \mathcal{E}_\tau, \mathcal{Q}, \mathcal{R}, \phi, \psi), \forall \tau \in \mathcal{R}$, in which $\mathcal{V}$ is a set of vertices, $\mathcal{E}$ is a set of edges, $\phi$ is a vertex type mapping function $\phi : \mathcal{V} \to \mathcal{Q}$ and $\psi$ is an edge type mapping function $\psi : \mathcal{E} \to \mathcal{R}$. It is defined that $|\mathcal{Q}| > 1$ and $|\mathcal{R}| > 1$.

**Definition 4. Inductive Learning** For any type of graph-based model, inductive learning describes its ability to generalize to unseen nodes and edges based on patterns learned on the training graph. Thus, the ability of a model trained on $\mathcal{G}_{\text{train}} = (\mathcal{V}_{\text{train}}, \mathcal{E}_{\text{train}})$ to make inferences on a graph $\mathcal{G}_{\text{test}} = (\mathcal{V}_{\text{test}}, \mathcal{E}_{\text{test}})$ where $\mathcal{V}_{\text{train}} \bigcap \mathcal{V}_{\text{test}} = \mathcal{E}_{\text{train}} \bigcap \mathcal{E}_{\text{test}} = \emptyset$ (Hamilton (2020)).

## 2.1 THE HETEROGENEOUS MULTIPLEX GRAPH ENCODER

We outline the approach of maintaining the multiplex graph structure $\mathcal{G} = (\mathcal{V}, \mathcal{E}_\tau, \mathcal{Q}, \mathcal{R}, \phi, \psi), \forall \tau \in \mathcal{R}$ throughout the learning pipeline by training separate encoders for each multiplex layer $\tau \in \mathcal{R}$. Each layer's encoder is composed of neural message-passing operators $q \in \mathcal{Q}$, resulting in $|\mathcal{R}| \times |\mathcal{Q}|$ operators, each with their own aggregation and update operators, as defined in Equation (1).

$$
\begin{aligned}
\mathbf{h}_{u,\tau}^{(k)} &\leftarrow \text{UPDATE}_{k,q,\tau} \left( \mathbf{h}_{u,\tau}^{(k-1)}, \text{AGGREGATE}_{k,q,\tau} \left( \left\{ \mathbf{h}_{v,\tau}^{(k-1)}, \forall v \in \mathcal{N}_\tau(u) \right\} \right) \right) \\
&= \text{UPDATE}_{k,q,\tau} \left( \mathbf{h}_{u,\tau}^{(k-1)}, \mathbf{m}_{\mathcal{N}_\tau(u)}^{(k-1)} \right)
\end{aligned}
\tag{1}
$$

Here, $k \in \{1, 2, \cdots, K\}$ indicates the depth of the neighborhood samples being aggregated, where $K$ is the total number of layers in the neighborhood sampling tree. The node type $q \in \mathcal{Q}$ is determined by the node type mapping function $q = \phi(u)$, and the message, denoted by $\mathbf{m}_{\mathcal{N}_\tau(u)}^{(k-1)}$, is aggregated from the neighborhood of node $u$ in layer $\tau$.

This adaption of the `GraphSAGE` algorithm by Hamilton et al. (2017) for inductive representation learning is described in Algorithm 1. We define this encoding algorithm as the Heterogeneous Multiplex `GraphSAGE` (`HMGSAGE`) encoder. We note that a mini-batching procedure is used for training `HMGSAGE` as described by Hamilton et al. (2017).

## 2.2 THE HETEROGENEOUS MULTIPLEX GRAPH AUTOENCODER

In this section, a GAE model is defined for HMGs, which takes a set of embedding vectors $\mathbf{z}_{u,\tau}, \forall u \in \mathcal{V}, \tau \in \mathcal{R}$ generated by `HMGSAGE` as input. This model is trained to reconstruct an input graph by estimating both the probability of an edge's existence, and the similarity between all pairs of nodes for each layer $\mathbf{S}_\tau[u, v], \forall \tau \in \mathcal{R}$. A variety of similarity metrics can be selected, each of which captures relationships between nodes in a different manner.

To construct this GAE, we define two sets of decoders. First, a layer-wise set of reconstruction decoders $\text{DEC}_{\tau,\text{neg}}, \forall \tau \in \mathcal{R}$, each of which outputs the probability of an edge existing between two nodes within its layer $\tau \in \mathcal{R}$. A second set of decoders $\text{DEC}_{\tau,\text{sim}}, \forall \tau \in \mathcal{R}$ reconstructs each layers' similarity matrix, $\hat{\mathbf{S}}_\tau$.

The GAE and reconstruction decoders $\text{DEC}_{\tau,\text{sim}}, \forall \tau \in \mathcal{R}$ are trained with a noise contrastive approach, in which reconstruction error is calculated using cross-entropy loss with negative sampling

---

**Algorithm 1** `HMGSAGE` encoder forward propagation

---

**Input**:
    Heterogeneous Multiplex Graph $\mathcal{G}(\mathcal{V}, \mathcal{E}_\tau, \mathcal{R}, \mathcal{Q}, \phi, \psi)$, $\forall\, \tau \in \mathcal{R}$;
    Input features $\{\mathbf{x}_u, \forall\, u \in \mathcal{V}\}$;
    Sampling height $K$;
    Update functions $\text{UPDATE}_{k,q,\tau}$, $\forall\, k \in \{1, \cdots, K\}$, $\tau \in \mathcal{R}, q \in \mathcal{Q}$;
    Aggregator functions $\text{AGGREGATE}_{k,q,\tau}$, $\forall\, k \in \{1, \cdots, K\}$, $\tau \in \mathcal{R}, q \in \mathcal{Q}$;
    Neighbourhood function $\mathcal{N}_\tau : v \to 2^v$, $\forall\, \tau \in \mathcal{R}$;
**Output**: Low-dimensional node embedding vectors $\mathbf{z}_{u,\tau}$, $\forall\, u \in \mathcal{V}, \tau \in \mathcal{R}$

1:  **procedure** $\text{HMGSAGE}(\mathcal{G}, \mathbf{x}_v; K, \text{UPDATE}_{k,q,\tau}, \text{AGGREGATE}_{k,q,\tau}, \mathcal{N}_\tau)$
2:     **for** $\tau \in \mathcal{R}$ **do**
3:         $\mathbf{h}_{u,\tau}^{(0)} \leftarrow \mathbf{x}_u$, $\forall\, u \in \mathcal{V}$
4:         **for** $k = 1 \cdots K$ **do**
5:             **for** $u \in \mathcal{V}$ **do**
6:                 $q \leftarrow \phi(u)$
7:                 $\mathbf{m}_{\mathcal{N}_\tau(u)}^{(k-1)} \leftarrow \text{AGGREGATE}_{k,q,\tau}\left(\left\{\mathbf{h}_{v,\tau}^{(k-1)}, \forall v \in \mathcal{N}_\tau(u)\right\}\right)$
8:                 $\mathbf{h}_{u,\tau}^{(k)} = \text{UPDATE}_{k,q,\tau}\left(\mathbf{h}_{u,\tau}^{(k-1)}, \mathbf{m}_{\mathcal{N}_\tau(u)}^{(k-1)}\right)$
9:                 $\mathbf{h}_{u,\tau}^{(k)} \leftarrow \mathbf{h}_{u,\tau}^{(k)}/||\mathbf{h}_{u,\tau}^{(k)}||_2$
10:            **end for**
11:         **end for**
12:         $\mathbf{z}_{u,\tau} \leftarrow \mathbf{h}_{u,\tau}^{K}$, $\forall u \in \mathcal{V}$
13:     **end for**
14: **end procedure**

---

(Grover & Leskovec (2016)). This can be computed using a Monte Carlo approach by sampling negative edge examples during training as shown in Equation (2),

$$\mathcal{L}_{\tau,\text{neg}} = \sum_{(u,v)\in\mathcal{E}_\tau} \left( -\log\left(\sigma\left(\text{DEC}_{\tau,\text{neg}}(\mathbf{z}_{u,\tau}, \mathbf{z}_{v,\tau})\right)\right) - \sum_{v_n \in \mathcal{P}_{n,u}} \left[\log\left(\sigma\left(-\text{DEC}_{\tau,\text{neg}}\left(\mathbf{z}_{u,\tau}, \mathbf{z}_{v_n,\tau}\right)\right)\right)\right] \right) \tag{2}$$

where $\mathcal{P}_{n,u}$ is the set of negative edges obtained by sampling $v_n \sim \mathcal{U}\left(\mathcal{V} \setminus \mathcal{N}_\tau(u)\right)$ and $\sigma$ is the logistic function. The cardinality of the negative samples, $|\mathcal{P}_{n,u}|$, is a hyperparameter, typically assigned a low value.

The second loss term used is the Mean Squared Error (MSE) reconstruction loss. For each layer $\tau \in \mathcal{R}$, the similarity measure $\mathbf{S}_\tau[u,v]$ is defined as the edge weights $\mathbf{A}_\tau[u,v]$. This loss is calculated according to Equation (3).

$$\mathcal{L}_{\tau,\text{sim}} = \frac{1}{|\mathcal{E}_\tau|} \sum_{(u,v)\in\mathcal{E}_\tau} \left(\mathbf{A}_\tau[u,v] - \text{DEC}_{\tau,\text{sim}}(\mathbf{z}_{u,\tau}, \mathbf{z}_{v,\tau})\right)^2 \tag{3}$$

The final loss is calculated as a weighted sum of the two, as per Equation (4),

$$\mathcal{L}_\tau = \lambda\mathcal{L}_{\tau,\text{neg}} + (1-\lambda)\mathcal{L}_{\tau,\text{sim}} \tag{4}$$

where the weight $\lambda$ is a hyperparameter defined in Appendix D.

The final step aggregates the individual layer losses, while applying a weighting factor $\propto \frac{1}{|\mathcal{E}_\tau|}$ to improve model performance on imbalanced multiplex graphs (See Appendix E.3).

$$\mathcal{L} = \sum_{\tau\in\mathcal{R}} w_\tau \mathcal{L}_\tau,$$

where $w_\tau$ is the edge weighting factor. The GAE extending the `HMGSAGE` encoder is illustrated in Figure 1, and an algorithmic description of the model's forward pass is given in Appendix B. This model is trained using stochastic gradient descent (Kingma & Ba (2015)).

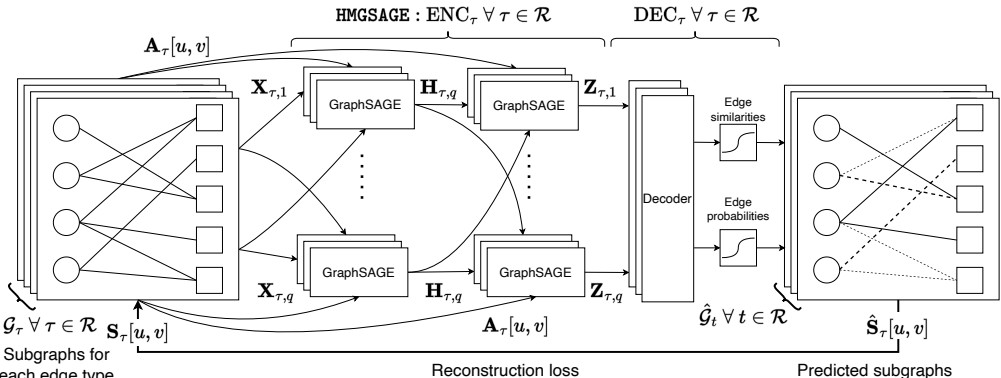

Figure 1: GAE model architecture

## 2.3 SUPERVISED REPRESENTATION LEARNING FOR INDUCTIVE HMG NODE CLASSIFICATION

We introduce the SECSAGE model, designed to perform node classification on Heterogeneous Multiplex Graphs. The SECSAGE architecture uses SECO to simultaneously train an HMGSAGE encoder and layer-wise classifiers $\text{CLASS}_\tau$, $\forall\,\tau\in\mathcal{R}$. At each layer $\tau\in\mathcal{R}$, the encoder generates embeddings $\mathbf{z}_{u,\tau}$, $\forall\,u\in\mathcal{V}$, which are used as inputs to $\text{CLASS}_\tau$. The classifier applies a linear transformation to the embeddings, then passes them through a softmax layer to calculate the predicted class probabilities $\hat{\mathbf{y}}_{u,\tau}$. A detailed algorithm for the forward pass of the SECSAGE decoder is given in Appendix B. Loss is computed using negative log-likelihood, as shown in Equation 5,

$$\mathcal{L}_{\tau,\text{CLASS}_\tau} = -\frac{1}{|\mathcal{V}|} \sum_{u\in\mathcal{V}} \sum_{c=1}^{C} y_{u,c} \log\left(\hat{y}_{u,\tau,c}\right) \tag{5}$$

where $\mathbf{y}_u$ and $\hat{\mathbf{y}}_{u,\tau}$ are the one-hot encoded class labels and class probabilities for node $u$ layer $\tau$, respectively. $C$ is the number of classes present in the dataset. The final loss is calculated by aggregating the classification loss weighted for each multiplex layer $w_\tau \,\forall\,\tau\in\mathcal{R}$, given by Equation (6)

$$\mathcal{L}_{\text{CLASS}} = \sum_{\tau\in\mathcal{R}} w_\tau \mathcal{L}_{\tau,\text{CLASS}}. \tag{6}$$

During prediction, $\hat{\mathbf{y}}_{u,\tau}$ is fused using soft (mean) or hard fusion across the $\mathcal{R}$ dimension. A schematic of the HMG node classification model is shown in Figure 2.

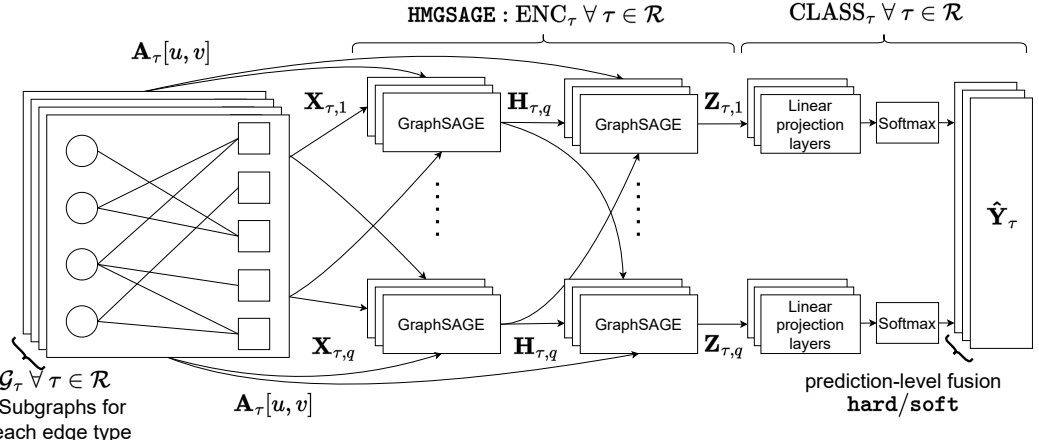

Figure 2: HMG node classification model

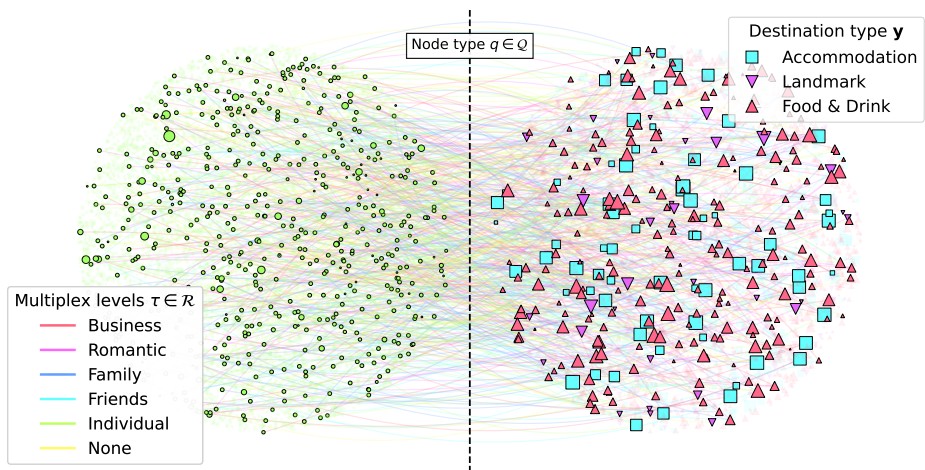

Figure 3: Visualization of the "Dubai Travel" dataset used for the experiments. Nodes on the left belong the "Traveller" node type while nodes on the right belong to the "Destination" node type. The sizes of the nodes are proportional to their degree.

## 3 EXPERIMENTS

In this section, we introduce a real-world dataset, "Travel Dubai", and use it to compare the performance of our proposed models to that of baselines. We also perform experiments on the "Amazon" product rating dataset described by Hou et al. (2024). We first present results highlighting the advantages of preserving the structure of an HMG throughout the training of a GAE, comparing the GAE model proposed in Section 2.2 to that of Model C1 on both the "Travel Dubai" and "Amazon" datasets. Afterwards, we show that SECSAGE outperforms Models C1, C2, and C3 on node classification tasks, showing the merits of Simultaneous Encoder-Classifier Optimization on both datasets.

### 3.1 DATA

This paper introduces a travel review dataset collected from proprietary sources based on public travel and tourism data. The data includes a travel network comprised of "Traveller" and "Destination" nodes, i.e., $\mathcal{Q} = \{\text{Traveller}, \text{Destination}\}$. These are connected by multiplex layers defined as $\mathcal{R} = \{\text{Business}, \text{Romantic}, \text{Group}, \text{Family}, \text{Individual}, \text{None}\}$, where unlabeled edges are assigned to None. The structure of the raw data is described in Appendix C.1.

We also use the "Amazon" dataset which is much larger in size relative to "Travel Dubai". It has two node types "Users" and "Items" connected by reviews in multiplex layers $\mathcal{R} = \{\text{Verified purchase}, \text{Unverified purchase}\}$. See Appendix C.2 for details.

Both datasets are processed into HMGs, and all nodes and edges are assigned to their node types and multiplex layers, respectively. Each edge is weighted according to its review rating, a continuous score between 1 and 5, where 5 is a positive rating. A downsampled version of the "Travel Dubai" graph is shown in Figure 3.

The HMGs are split into distinct train and test subgraphs $\mathcal{G}_{\text{train}}$ and $\mathcal{G}_{\text{test}}$ using a 70% - 30% split on the graph nodes. We ensure that these subgraphs have mutually exclusive node and edge sets. We also stratify nodes and edges by their respective types, to ensure an equivalent distribution in both the train and test subgraphs (see Appendix C for further details).

To test the effectiveness of graph-level information fusion, we created flattened versions of both subgraphs, $\mathcal{G}_{\text{train}}$ and $\mathcal{G}_{\text{test}}$ by simply collapsing all multiplex layers into one graph, i.e., $\mathcal{G}(\mathcal{V}, \mathcal{E}_\tau)$, $\forall \tau \in \mathcal{R} \rightarrow \mathcal{G}_{\text{flattened}}(\mathcal{V}, \mathcal{E}_{\text{total}})$, where $\mathcal{E}_{\text{total}} = \mathcal{E}_1 \cup \mathcal{E}_2 \cup \cdots \cup \mathcal{E}_{|\mathcal{R}|}$.

## 3.2 GAE TRAINING AND PERFORMANCE

Using the GAE defined in Section 2.2, we train the `HMGSAGE` encoder on the train subgraph $\mathcal{G}_{\text{train}}$ for both the edge prediction and edge rating tasks. We use the aggregate loss defined in Equation (4). Afterwards, we evaluate the trained model on the test subgraph $\mathcal{G}_{\text{test}}$. We also train Model C1, a GAE with a standard `GraphSAGE` backbone, on the flattened subgraphs. The hyperparameters of our GAE models are reported in Appendix D.

To evaluate the performance of the GAEs on $\mathcal{G}_{\text{test}}$, we scale all edge weights to lie on the range [0,1]. Next, we binarize the edge weights, converting them into binary (0 or 1) values based on a predefined threshold which we define as the expectation.

$$\mathbf{A}_{\tau,\text{binary}}[u,v] = \left\{ \begin{array}{ll} 0, & \text{for } \mathbf{A}_{\tau,\text{binary}}[u,v] < \mathbb{E}(\{\mathbf{A}_{\tau}[u,v], \ \forall \ u, v \in \mathcal{E}_{\tau}\}) \\ 1, & \text{for } \mathbf{A}_{\tau,\text{binary}}[u,v] \geq \mathbb{E}(\{\mathbf{A}_{\tau}[u,v], \ \forall \ u, v \in \mathcal{E}\tau\}) \end{array} \right. \tag{7}$$

We compare $\mathbf{A}_{\tau,\text{binary}}[u,v]$ with $\hat{\mathbf{A}}_{\tau,\text{binary}}[u,v]$ to compute the area under the curve (AUC) of the Receiver Operating Characteristic (ROC) curve. We also check the AUC of the Precision-Recall (PR-RC) curves for the adjacency prediction task which provides more informative metrics for imbalanced data, as is the case for $\mathbf{A}[u,v]$ in both datasets (see Appendix C for detailed statistics of the datasets). We also compute the ROC AUC for the edge prediction task, $\text{DEC}_{\tau,\text{neg}}(\mathbf{z}_{u,\tau}, \mathbf{z}_{v,\tau})$. Finally, we compute accuracy and F1 scores, based on probability thresholds selected in Appendices E.1 and E.2 for the "Travel Dubai" and "Amazon" datasets, respectively. Table 2 summarizes all relevant metrics for each GAE implementation. The results show that Graph-level fusion reduced

Table 2: Binary classification results for the GAE models

| Decoder task | AUC ROC | AUC PR-RC | Accuracy | F1-score | Threshold |
|---|---|---|---|---|---|
| **"Travel Dubai" dataset** | | | | | |
| *With* graph-level fusion, `GraphSAGE` encoder-decoder | | | | | |
| Edge prediction | 0.88969 | 0.85822 | 0.84466 | 0.865546 | 0.04654 |
| Edge rating | 0.81785 | 0.91038 | 0.83441 | **0.894465** | 0.57059 |
| *Without* graph-level fusion, `HMGSAGE` encoder-decoder | | | | | |
| Edge prediction | **0.90186** | **0.86096** | **0.85437** | **0.866535** | 0.00358 |
| Edge rating | **0.89921** | **0.95200** | **0.83711** | 0.891601 | 0.66482 |
| **"Amazon" dataset** | | | | | |
| *With* graph-level fusion, `GraphSAGE` encoder-decoder | | | | | |
| Edge prediction | 0.96690 | 0.92758 | 0.89065 | **0.863585** | 0.02272 |
| Edge rating | 0.96567 | 0.97893 | 0.90825 | 0.927288 | 0.78718 |
| *Without* graph-level fusion, `HMGSAGE` encoder-decoder | | | | | |
| Edge prediction | **0.96733** | **0.92889** | **0.89126** | 0.863171 | 0.02311 |
| Edge rating | **0.97435** | **0.98453** | **0.91668** | **0.935017** | 0.78496 |

the performance of the model on edge prediction tasks in both datasets, suggesting that better and more expressive representations are learned on HMG-based models.

## 3.3 CLASSIFIER TRAINING AND PERFORMANCE

We train `SECSAGE` on the train subgraph $\mathcal{G}_{\text{train}}$ for a node classification task to predict class membership of "Destination" and "Item" nodes for the "Travel Dubai" and "Amazon" datasets, respectively. The labels are {"Landmark", "Accommodation", "Food & Drink"} and {"Automotive", "Beauty", "Lawn, Garden & Patio"} for the "Travel Dubai" and "Amazon" datasets, respectively. The classifier is trained using the loss defined in Equation (6). We evaluate the trained model on the test subgraph $\mathcal{G}_{\text{test}}$ and compare it to models C1, C2, and C3. The hyperparameters of our `SECSAGE` model are reported in Appendix D.

To evaluate the performance of the `SECSAGE` model on $\mathcal{G}_{\text{test}}$, we calculate each class-wise F1 score, and compute the mean to obtain a macro-averaged F1 score for the model. This places equal weight

on each class, providing a more balanced view of the model's performance since the classes are imbalanced in both datasets (See Appendix C). We also calculate the raw accuracy of the model, which is the proportion of correctly classified nodes to the total number of nodes in the test sub-graph. The results are summarized in Table 3. We also present the confusion matrix for the models trained with and without SECO in Appendices E.1 and E.1 for the "Travel Dubai" and "Amazon" datasets, respectively. The results show that using SECO to simultaneously train the encoder and

Table 3: Node classification results comparing models C1, C2, C3, and `SECSAGE` on all datasets

| **Prediction-fusion** | - | - | `soft` | `hard` | `soft` | `hard` |
|---|---|---|---|---|---|---|
| **Graph-fusion** | ✓ | ✓ | ✗ | ✗ | ✗ | ✗ |
| with SECO | ✗ | ✓ | ✗ | ✗ | ✓ | ✓ |
| **Model** | C1 | C2 | C3 | C3 | `SECSAGE` | `SECSAGE` |
| **"Travel Dubai" dataset** | | | | | | |
| Accuracy | 0.61491 | 0.72727 | 0.74157 | 0.74566 | 0.74566 | **0.75383** |
| Micro F1 | 0.59823 | 0.74893 | 0.76521 | 0.76855 | 0.76886 | **0.77615** |
| **Macro F1** | 0.36606 | 0.67659 | 0.69267 | 0.69407 | 0.69648 | **0.70127** |
| **"Amazon" dataset** | | | | | | |
| Accuracy | 0.19676 | 0.18713 | 0.36424 | 0.28898 | 0.44405 | **0.44516** |
| Micro F1 | 0.18380 | 0.17266 | 0.41348 | 0.32382 | **0.48638** | 0.48066 |
| **Macro F1** | 0.20814 | 0.18240 | 0.32071 | 0.26037 | **0.37059** | 0.35510 |

downstream classifier improves its performance in terms of macro-averaged F1 score. We visualize the embeddings learned by the `GraphSAGE` encoder on the "Travel Dubai" dataset with and without SECO (C1 versus C2) in Figure 4. The embeddings are projected into 2 dimensions using t-Distributed Stochastic Neighbor Embedding (t-SNE). We observe a larger degree of separation between the classes when using SECO compared to the baseline model as given by the silhouette score. This can help explain the improved classification performance of SECO, due to the better separation of the classes in the embedding domain and subsequent improved decision boundaries learned by the downstream classifier. We also show in Appendix D that the computational time needed to train the models slightly increases when using parallel `GraphSAGE` layers in `HMGSAGE` and `SECSAGE` relative to their flattened counterparts. However, SECO is shown to reduce the time needed to learn embeddings, as the encoder is simultaneously trained without the need of a GAE or other contrastive approaches.

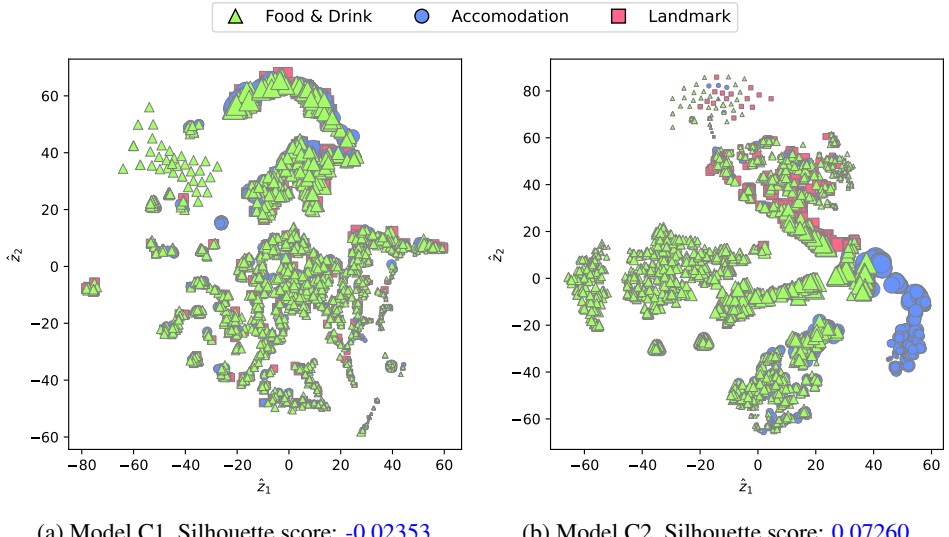

(a) Model C1, Silhouette score: -0.02353      (b) Model C2, Silhouette score: 0.07260

Figure 4: Two-dimensional projection of $\mathbf{z}_u$, $\forall \phi(u) =$ "Destination" in the "Travel Dubai" dataset for Models (a) C1 and (b) C2. The sizes of the points are proportional to the node degree.

## 4 CONCLUSION

In this paper, we propose three GRL models and evaluate them on a new real-world dataset and a large benchmark dataset. First, we introduce an encoder for Heterogeneous Multiplex Graphs denoted `HMGSAGE`. Next, we extend `HMGSAGE` to create a Graph Autoencoder which performs inductive edge prediction on HMG-structured data. Finally, we introduce the topic of Simultaneous Encoder-Classifier Optimization. We use it to extend `HMGSAGE` for the task of inductive node classification on HMGs, creating a model denoted `SECSAGE`. After comparing these to baseline models, we show that our architectures outperform those using other information fusion techniques on both edge prediction and node classification tasks.

From these comparisons, two primary conclusions can be drawn. We show that preserving the graph structure of an HMG early in the learning pipeline leads to the learning of better embeddings. This is evidenced by the performance of our GAE using `HMGSAGE` against a similar model using `GraphSAGE` and graph-level fusion. We also show that by using SECO, models learn task-optimal embeddings, improving the performance of a particular downstream task. This improves even further when combined with prediction-level fusion as opposed to graph-level.

We also note that multiplex layers are often imbalanced in real-world datasets, and can be handled by using weighted loss terms in combination with dedicated GRL operators for each layer as demonstrated in this paper. Furthermore, edge weights and class labels can also be imbalanced, and appropriate metrics should be used for evaluating model performance. This offers some practical considerations when working with graph-based recommendation systems.

There are several possibilities for further research towards better GRL on HMGs. More testing should be conducted into measuring the performance of `SECSAGE` on other HMG datasets, with comparisons against more baselines using other encoders such as DGI interchangeably with `HMGSAGE`. Additionally, the benefits of SECO should be studied on models using other types of information fusion, specifically GNN-level and embedding-level. The potential to generalize SECO to produce task-optimal embeddings for other tasks such as clustering, graph-level classification, and combinations thereof also merit investigation.

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

# A   BACKGROUND

This section provides a background on related concepts used throughout the paper.

**Graph Autoencoder (GAE)**   A Graph Autoencoder is a neural-network-based machine learning model designed to learn low-dimensional latent representations, known as embedding vectors, of various elements of graph-structured data. In this paper we focus on learning embeddings for the different nodes in our graph $\mathbf{z}_u, \forall u \in \mathcal{V}$. We use an encoder-decoder paradigm similar to the seminal work of Kipf & Welling (2016). The *encoder* is constructed using several GNN operators and maps nodes to high-dimensional vector space $\mathrm{ENC} : \mathcal{V} \mapsto \mathbb{R}^d$, where $d$ is the dimensionality of the embedding vectors $\mathbf{z}_v \in \mathbb{R}^d$ for $v \in \mathcal{V}$.

$$\mathrm{ENC}(v) = \mathbf{Z}[v], \tag{8}$$

The *decoder* is a model that obtains the embeddings $\mathbf{z}_u$ in the low-dimensional space and reconstructs information about the nodes' local neighborhood in the original graph. As an example, the reconstructed information could represent the set of neighbours of node $u$, $\mathcal{N}(u)$ or its row $\mathbf{A}[v]$ in the adjacency matrix. However, a better approach is to use a pairwise decoder which reconstructs information about the relationship between pairs of nodes, i.e.,

$$\mathrm{DEC}(\mathrm{ENC}(u), \mathrm{ENC}(v)) = \mathrm{DEC}(\mathbf{z}_u, \mathbf{z}_v) \approx \mathbf{S}[u, v], \tag{9}$$

where $\mathbf{S}[u, v]$ is the reconstruction objective, e.g., $\mathbf{S}[u, v] \triangleq \mathbf{A}[u, v]$, although different objectives can be defined other than the adjacency matrix.

Finally, a reconstruction loss can be defined over a set of training nodes $\mathcal{D} \in \mathcal{V}$ and used to optimize the parameters of the encoder and decoder using gradient descent:

$$\mathcal{L} = \sum_{(u,v) \in \mathcal{D}} \ell(\mathrm{DEC}(\mathbf{z}_u, \mathbf{z}_v), \mathbf{S}[u, v]),$$

where $\ell : \mathbb{R} \times \mathbb{R} \mapsto \mathbb{R}$ is a loss function such as mean-squared error or binary cross entropy loss.

There are many different models for the decoder, the similarity measure $\mathbf{S}[u, v]$, and loss function, different combinations of which results in different representation learning algorithms. In this paper, we focus on a simple decoder that uses a feedforward layer to concatenate pairs of node embeddings following by a sigmoid layer to scale the predicted similarity measure between 0 and 1 as explained in Section 2.2

# B ALGORITHMS FOR DOWNSTREAM TASKS

In this section, we provide the detailed algorithmic descriptions for the forward pass of the GAE model in Section 2.2, and the supervised learning task for HMG node classification in Section 2.3. The GAE model is described in Algorithm 2 below and is used to predict the probability matrix $\hat{\mathbf{P}}$ which gives the probability of an edge between pairs of nodes. The adjacency matrix $\hat{\mathbf{A}}$ is also predicted for each layer $\tau \in \mathcal{R}$ during the forward pass. We assume the node embeddings $\mathbf{z}_{u,\tau}$ are already computed using the `HMGSAGE` encoder.

---

**Algorithm 2** HMG GAE forward propagation

---

    **Input**:
        The node embeddings $\mathbf{z}_{u,\tau}$, $\forall\, u \in \mathcal{V}, \tau \in \mathcal{R}$;
        The weights matrices $\mathbf{W}_{\tau,\mathrm{neg}}$, $\forall\, \tau \in \mathcal{R}$, $\mathbf{W}_{\tau,\mathrm{sim}}$, $\forall\, \tau \in \mathcal{R}$;
        The non-linearity function $\sigma$;
        The adjacency matrix $\mathbf{A}_\tau$, $\forall\, \tau \in \mathcal{R}$;
    **Output**: The similarity matrix $\hat{\mathbf{P}}_\tau \in \mathbb{R}^{|\mathcal{E}|}$, $\forall\, \tau \in \mathcal{R}$; $\hat{\mathbf{A}}_\tau \in \mathbb{R}^{|\mathcal{E}|}$, $\forall\, \tau \in \mathcal{R}$
1:  **procedure** HMGGAE($\mathbf{z}_{u,\tau}, \mathbf{z}_{v,\tau}, \mathbf{W}_{\tau,\mathrm{neg}}, \mathbf{W}_{\tau,\mathrm{sim}}$, $\forall\, \mathcal{R}, \sigma$)
2:     **for** $\tau \in \mathcal{R}$ **do**
3:         $\hat{\mathbf{P}}_\tau[u, v] \leftarrow \sigma\left(\mathbf{W}_{\tau,\mathrm{neg}} \cdot \mathrm{CONCAT}\left(\mathbf{z}_{u,\tau}, \mathbf{z}_{v,\tau}\right)\right)$
4:         $\hat{\mathbf{A}}_\tau[u, v] \leftarrow \sigma\left(\mathbf{W}_{\tau,\mathrm{sim}} \cdot \mathrm{CONCAT}\left(\mathbf{z}_{u,\tau}, \mathbf{z}_{v,\tau}\right)\right)$
5:     **end for**
6:  **end procedure**

---

The supervised learning task for HMG node classification is also described in Algorithm 3 below. The model is used to predict the class probabilities $\hat{\mathbf{y}}$ for each node $u \in \mathcal{V}$ given the node embeddings $\mathbf{z}_{u,\tau}$ that are computed using the `HMGSAGE` encoder.

The model consists of a linear layer followed by a softmax layer to predict the class probabilities. The linear layer maps the node embeddings to the class space using the weights matrices $\mathbf{W}_\tau \in \mathbb{R}^{C \times d}$, where $C$ is the number of node labels and $d$ is the dimensionality of the node embeddings.

---

**Algorithm 3** Supervised learning for HMG node classification

---

    **Input**:
        The node embeddings $\mathbf{z}_{u,\tau}$, $\forall\, u \in \mathcal{V}, \tau \in \mathcal{R}$;
        The number of node labels $C$;
        The projection layer weight matrices $\mathbf{W}_\tau \in \mathbb{R}^{C \times d}$, $\forall\, \tau \in \mathcal{R}$;
    **Output**: The predicted class probabilities $\hat{\mathbf{y}}_{u,\tau}$, $\forall\, u \in \mathcal{V}$, $\forall\, \tau \in \mathcal{R}$
1:  **procedure** HMGCLASS($\mathbf{z}_{u,\tau}, \mathbf{W}_\tau$, $\forall\, \mathcal{R}, C$)
2:     **for** $\tau \in \mathcal{R}$ **do**
3:         **for** $c = 1$ to $C$ **do**
4:             $\hat{y}_{u,\tau,c} \leftarrow \dfrac{\exp\left(\mathbf{W}_\tau \mathbf{z}_{u,\tau}\right)_c}{\sum_{c'=1}^{C} \exp\left(\mathbf{W}_\tau \mathbf{z}_{v,\tau}\right)_{c'}}, \ \forall\, c \in \mathcal{C}$
5:         **end for**
6:     **end for**
7:  **end procedure**

---

# C EXPERIMENTAL DATASET DETAILS

## C.1 "TRAVEL DUBAI" DATASET

We first describe the statistics of the "Travel Dubai" dataset which we hope to publish and add to existing benchmark datasets popular in the field of Network Science and GRL. The Entity-Relationship (ER) diagram for the raw data before processing it into an HMG is shown in Figure 5. The HMG

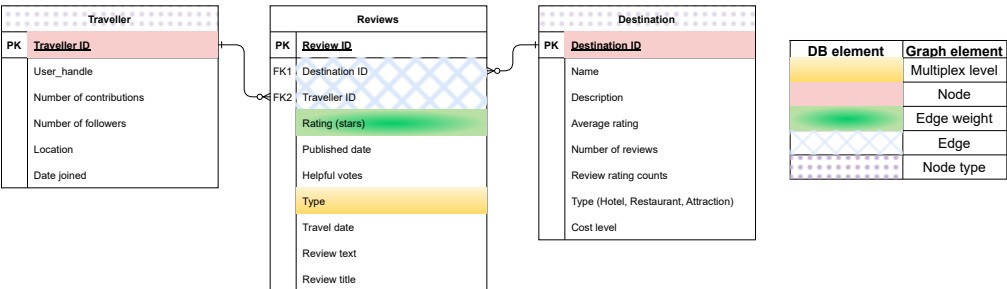

Figure 5: ER diagram of raw "Dubai Travel" dataset used for algorithm evaluation.

constructed from this data is reported in the main body of the paper and a subset is plotted in Figure 3. Figure 6a shows the relative distribution of different "Destination" node labels (used for classification). We note that within the "Destination" nodes, the "Landmark" and "Accommodation" labels are underrepresented relative to "Food & Drink". Figure 6b shows a somewhat balanced distribution among the various multiplex layers. The HMG formed by this data consists

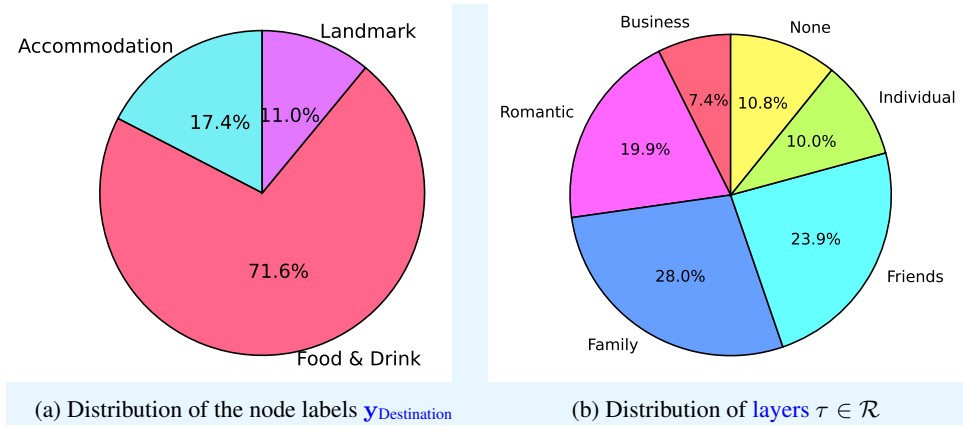

(a) Distribution of the node labels $\mathbf{y}_{\text{Destination}}$    (b) Distribution of layers $\tau \in \mathcal{R}$

Figure 6: Distribution of (a) the "Destination" node labels $\mathbf{y}_{\text{Destination}}$ and (b) edges within each multiplex layer of the "Dubai Travel" dataset.

of 20,969 nodes and 20,904 edges. The HMG is split into train and test subgraphs that are mutually exclusive, i.e., $\mathcal{V}_{\text{train}} \cup \mathcal{V}_{\text{test}} = \emptyset$ and $\mathcal{E}_{\tau,\text{train}} \cup \mathcal{E}_{\tau,\text{test}} = \emptyset, \ \forall \ \tau \in \mathcal{R}$. To achieve such a split, we performed a train-test split on the set of nodes $\mathcal{V}$ stratified by the node type $q \in \mathcal{Q}$. We then constructed the set of train edges by selecting all edges spanned by the train nodes, i.e., $\mathcal{E}_{\tau,\text{train}} = \{u, v \in \mathcal{E} | u \in \mathcal{V}_{\text{train}}, v \in \mathcal{V}_{\text{train}}, \forall \tau \in \mathcal{R}\}$.

We show the distribution of the marginal edge weights (given by a rating on a scale of 1 to 5) and distributed across different layers of the train and test HMGs in Figure 7. We can see that our train/test split procedure maintained the edge weight distribution in both graphs. Further, we can notice that extremely positive ratings (5) are far more frequent relative to other ratings justifying the use of metrics which penalize more on imbalanced data such as precision and recall as was done in Section 3.2. Furthermore, we show the relative distribution of all the edge types and nodes types in the complete, train, and test subgraphs is reported in Table 4. This shows that all datasets maintain

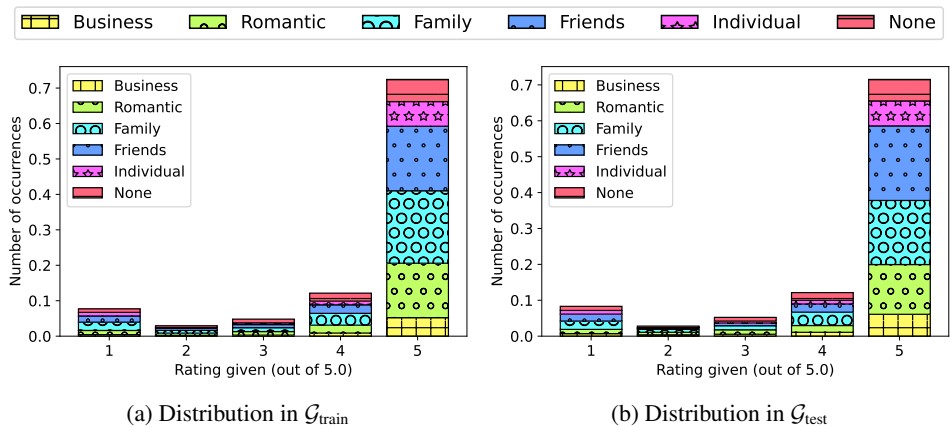

(a) Distribution in $\mathcal{G}_{\text{train}}$         (b) Distribution in $\mathcal{G}_{\text{test}}$

Figure 7: Probability Mass Function of the edge weights of the "Travel Dubai" multiplex graph $\mathbf{A}_\tau$, $\forall \tau \in \mathcal{R}$. The horizontal partitioning shows the distribution across the multiplex layers $\tau \in \mathcal{R}$ for (a) the train subgraph and (b) the test subgraph.

the same relative proportions across train/test splits. Finally, we report the absolute number of nodes and edges in each of the train, test, and full graphs in Table 5

Table 4: Statistics of train/test subgraphs of the "Travel Dubai" dataset

| Attribute | | Relative proportion | | |
| --- | --- | --- | --- | --- |
| | | Full graph $\mathcal{G}$ | Train graph $\mathcal{G}_{\text{train}}$ | Test graph $\mathcal{G}_{\text{train}}$ |
| Traveller | | 0.84444 | 0.84446 | 0.84438 |
| Destination | Landmark | 0.01707 | 0.01744 | 0.01621 |
| | Accommodation | 0.02709 | 0.02698 | 0.02734 |
| | Food & Drink | 0.11140 | 0.11112 | 0.11206 |
| Edge types $\tau \in \mathcal{R}$ | Business | 0.07391 | 0.07077 | 0.08792 |
| | Romantic | 0.19862 | 0.20401 | 0.18878 |
| | Family | 0.28023 | 0.27961 | 0.25836 |
| | Group | 0.23948 | 0.23806 | 0.26052 |
| | None | 0.10821 | 0.11015 | 0.10734 |
| | Individual | 0.09955 | 0.09741 | 0.09709 |

Table 5: Node and edge counts in train, test, and full graphs of the "Travel Dubai" dataset

| Attribute | | Relative proportion | | |
| --- | --- | --- | --- | --- |
| | | Full graph $\mathcal{G}$ | Train graph $\mathcal{G}_{\text{train}}$ | Test graph $\mathcal{G}_{\text{train}}$ |
| Traveller | | 17707 | 12395 | 5312 |
| Destination | Landmark | 358 | 256 | 102 |
| | Accommodation | 568 | 396 | 172 |
| | Food & Drink | 2336 | 1631 | 705 |
| Edge types $\tau \in \mathcal{R}$ | Business | 1545 | 717 | 163 |
| | Romantic | 4152 | 2067 | 350 |
| | Family | 5858 | 2833 | 479 |
| | Group | 5006 | 2412 | 483 |
| | None | 2262 | 1116 | 199 |
| | Individual | 2081 | 987 | 180 |

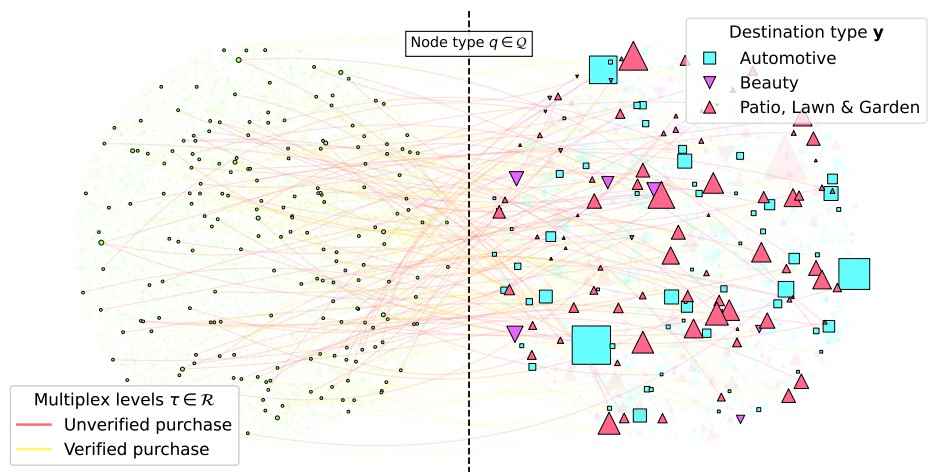

Figure 8: Visualization of the "Amazon" dataset used for the experiments. Nodes on the left belong the "User" node type while nodes on the right belong to the "Item" node type. The sizes of the nodes are proportional to their degree.

## C.2    "AMAZON" DATASET

In this section, we describe the details of the "Amazon" dataset reported by Hou et al. (2024). A subset is plotted in Figure 8. Figure 9a shows the relative distribution of different "Item" node labels (for classification). We note that within the "Item" nodes, the "Beauty" and "Patio, Lawn & Garden" labels are underrepresented relative to "Automotive". Figure 6b shows an extreme imbalance between the "Verified purchase" and "Unverified purchase" multiplex layers. The HMG for this

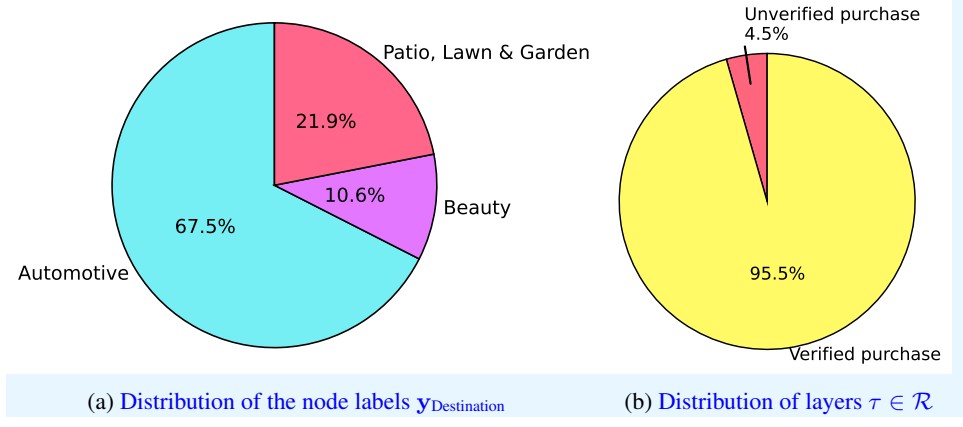

(a) Distribution of the node labels $\mathbf{y}_{\text{Destination}}$     (b) Distribution of layers $\tau \in \mathcal{R}$

Figure 9: Distribution of (a) the "Item" node labels $\mathbf{y}_{\text{Item}}$ and (b) edges within each multiplex layer of the "Amazon" dataset.

dataset consists of 1.51M nodes and 1.43M edges. Similar to the "Travel Dubai" dataset, The HMG is split into train and test subgraphs that are mutually exclusive, stratified by the node type $q \in \mathcal{Q}$.

We show the distribution of the marginal edge weights (given by a rating on a scale of 1 to 5) and distributed across different layers of the train and test HMGs in Figure 10. We note that positive ratings (5) are far more frequent relative to other ratings just as was observed with the "Travel Dubai" dataset. We also show the relative distribution of all the edge types and nodes types in the complete, train, and test subgraphs in Table 6. Finally, we report the absolute number of nodes and edges in each of the train, test, and full graphs in Table 7.

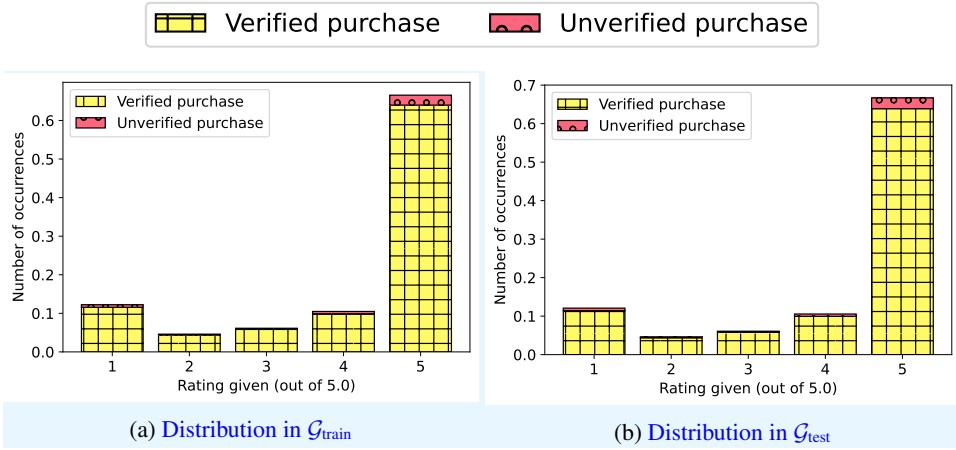

(a) Distribution in $\mathcal{G}_{\text{train}}$       (b) Distribution in $\mathcal{G}_{\text{test}}$

Figure 10: Probability Mass Function of the edge weights of the "Amazon" multiplex graph $\mathbf{A}_\tau$, $\forall \tau \in \mathcal{R}$. The horizontal partitioning shows the distribution across the multiplex layers $\tau \in \mathcal{R}$ for (a) the train subgraph and (b) the test subgraph.

Table 6: Statistics of train/test subgraphs of the "Amazon" dataset

| Attribute | | Relative proportion | | |
| --- | --- | --- | --- | --- |
| | | **Full graph** $\mathcal{G}$ | **Train graph** $\mathcal{G}_{\text{train}}$ | **Test graph** $\mathcal{G}_{\text{train}}$ |
| User | | 0.93689 | 0.93692 | 0.93691 |
| Item | Automotive | 0.04260 | 0.04260 | 0.004260 |
| | Beauty | 0.00666 | 0.006610 | 0.006787 |
| | Patio, Lawn & Garden | 0.01384 | 0.01388 | 0.01373 |
| Edge types $\tau \in \mathcal{R}$ | Verified purchase | 0.95548 | 0.95623 | 0.95406 |
| | Unverified purchase | 0.004452 | 0.004377 | 0.004564 |

Table 7: Node and edge counts in train, test, and full graphs of the "Amazon" dataset

| Attribute | | Relative proportion | | |
| --- | --- | --- | --- | --- |
| | | **Full graph** $\mathcal{G}$ | **Train graph** $\mathcal{G}_{\text{train}}$ | **Test graph** $\mathcal{G}_{\text{train}}$ |
| User | | 1.244M | 871K | 373.3K |
| Item | Automotive | 56.57K | 39.6K | 16.96K |
| | Beauty | 8.849K | 6.145K | 2.704K |
| | Patio, Lawn & Garden | 18.38K | 12.9K | 5.47K |
| Edge types $\tau \in \mathcal{R}$ | Verified purchase | 1.367M | 680.8K | 117.7K |
| | Unverified purchase | 63.7K | 31.16K | 5.667K |

## D    DETAILED TRAINING SETUP

In this section, we report the training settings used for all the studies in this paper. The hyper-paramters selected for each model developed in this paper are reported in Table 8.

Table 8: Hyperparameters used for training the GAE and classifiers

| Model | Parameter | Value | Description |
|---|---|---|---|
| HMGSAGE/ GraphSAGE encoders | $K$ | 3 | Sampling height for the `GraphSAGE` layers |
| | $\{|\mathcal{N}_1|, \cdots, |\mathcal{N}_K|\}$ | $\{25, 10, 5\}$ | Number of neighbourhood samples per sampling depth |
| | $|\mathcal{B}|$ | 1024 | Batch size for minibatch training of `GraphSAGE` layers |
| | $d$ | 32 | `HMGSAGE` embedding dimensionality |
| GAE decoder | $n_{\text{layers}}$ | 1 | Number of layers used by the concatenation decoder models |
| | $|\mathcal{P}_{u,n}|$ | 5 | Negative sampling frequency per node |
| | $\lambda$ | 0.5 | Weighting of edge prediction loss relative to similarity reconstruction loss |
| | $n_{\text{epochs}}$ | 2000 | Number of backpropagation steps |
| Classifier | $n_{\text{layers}}$ | 2 | Number of feedforward layers |
| | $h_{\text{dim}}$ | 128 | Intermediate dimension between layers |
| | $n_{\text{epochs}}$ | 2000 | Number of backpropagation steps |
| All models | $p_{\text{dropout}}$ | 0.1 | Dropout probability applied during back-propagation |

We also report the models' training performance and memory cost in terms of their training time and number of parameters, respectively in Table 9. We can see that the number of parameters grows linearly in the number of multiplex layers $|\mathcal{R}| = 5$ and $|\mathcal{R}| = 2$ for the "Travel Dubai" and "Amazon" datasets, respectively. The training time for the GAEs also increased in the absence of graph-level fusion, however the increase in training time was not as significant in both datasets since the total number of nodes and edges remains the same. Backpropagation of the training loss through the additional parameters resulted in this observed increase in training time. Another observation, is that the use of SECO also increased the training time (again due to the larger number of parameters being updated during backpropagation) albeit more noticeably in the larger "Amazon" dataset. The results in Table 9 suggest that the framework is scalable to larger datasets.

Table 9: Training performance and memory cost of all models reported in the paper. Training times are reported after 500 epochs have elapsed.

| Graph-fusion | ✗ | ✓ | ✓ | ✓ | ✗ | ✗ |
|---|---|---|---|---|---|---|
| with SECO | - | - | ✗ | ✓ | ✗ | ✓ |
| **Model** | GraphSAGE | HMGSAGE | C1 | C2 | C3 | SECSAGE |
| **"Travel Dubai" dataset** | | | | | | |
| Trainable parameters | 43330 | 259980 | 2307 | 28227 | 13842 | 169362 |
| Total parameters | 43330 | 259980 | 28227 | 28227 | 169362 | 169362 |
| Training time (s) | 113.306 | 230.742 | 22.1731 | 27.2747 | 45.2296 | 63.4619 |
| **"Amazon" dataset** | | | | | | |
| Trainable parameters | 42690 | 85380 | 2307 | 27587 | 4614 | 55174 |
| Total parameters | 42690 | 85380 | 27587 | 27587 | 55174 | 55174 |
| Training time (s) | 11204.7 | 13774 | 693.605 | 1502.15 | 806.7 | 1992.2 |

# E  DETAILED RESULTS

In this section, we provide detailed results for the training of the GAE models in Section 3.2.

## E.1  "TRAVEL DUBAI" DATASET RESULTS

We first report all the results for the models trained on the "Travel Dubai" dataset. Figures 11a and 11c show ROC and PR-RC curves, respectively for the edge prediction task. We also report the ROC and PR-RC curves for the edge rating task in Figures 11b and 11d, respectively. All figures show a comparison between the HMGSAGE and GraphSAGE encoders. For the edge prediction task, we

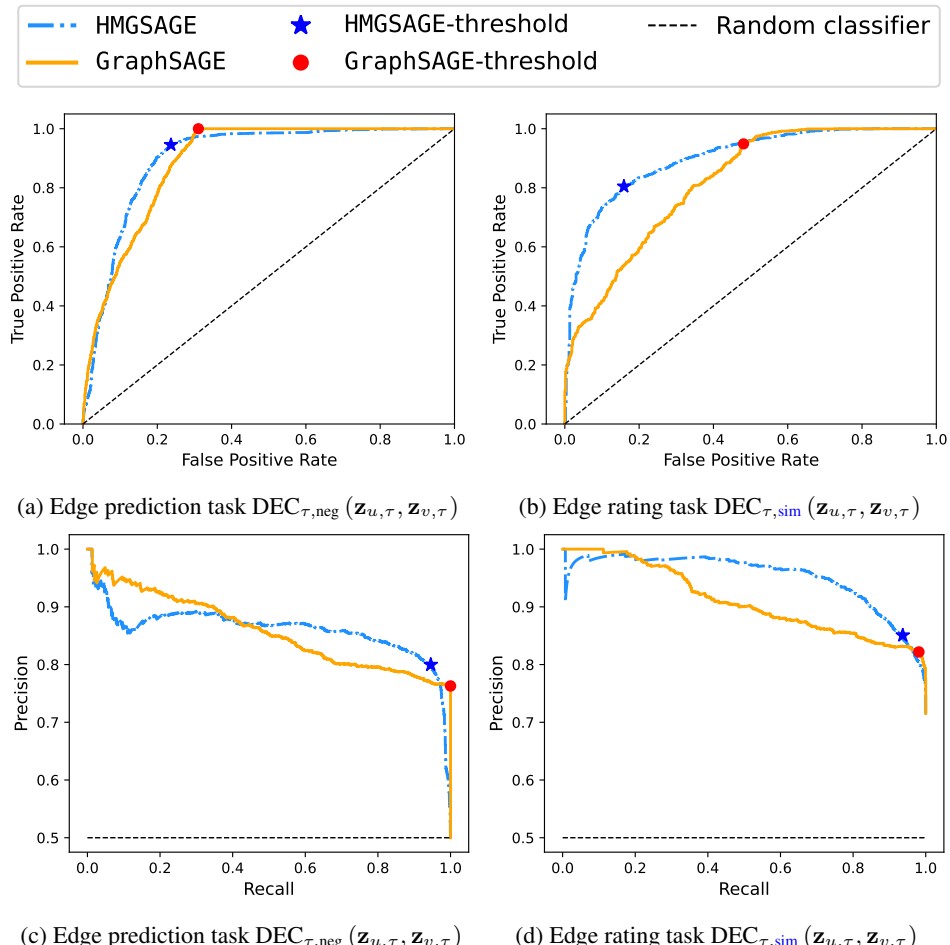

(a) Edge prediction task $\mathrm{DEC}_{\tau,\mathrm{neg}}\left(\mathbf{z}_{u,\tau}, \mathbf{z}_{v,\tau}\right)$      (b) Edge rating task $\mathrm{DEC}_{\tau,\mathrm{sim}}\left(\mathbf{z}_{u,\tau}, \mathbf{z}_{v,\tau}\right)$

(c) Edge prediction task $\mathrm{DEC}_{\tau,\mathrm{neg}}\left(\mathbf{z}_{u,\tau}, \mathbf{z}_{v,\tau}\right)$      (d) Edge rating task $\mathrm{DEC}_{\tau,\mathrm{sim}}\left(\mathbf{z}_{u,\tau}, \mathbf{z}_{v,\tau}\right)$

Figure 11: Detailed evaluation of the binary output of the GAE for the edge prediction and edge rating tasks of the "Travel Dubai" dataset using (a,b) ROC and (c,d) PR-RC curves. The notation HMGSAGE refers to the GAE model trained on the unflattened graph.

select a threshold value based on the ROC curve that maximizes the distance of the true positive rate and false positive rate to the diagonal line using Youden's index as shown in Figures 11a and 11b.

For the edge rating task, we select a threshold value based on the PR-RC curve that maximizes the F1 score as shown in Figures 11c and 11d. We report the values of the thresholds for the HMGSAGE and GraphSAGE models in Table 10.

Next, we visualize the embeddings learned by the HMGSAGE encoders in models C3 (without SECO) and SECSAGE (with SECO) in Figures 12 and 13, respectively. Note that for each node, a separate embedding is learned for each multiplex layer $\tau \in \mathcal{R}$. Comparing the embeddings of Model C3

Table 10: Threshold values for the GAE models trained on the "Travel Dubai" dataset. The bold values indicate the selected threshold values for each task.

| Decoder task | Threshold AUC ROC | Threshold PR-RC |
|---|---|---|
| *With* graph-level fusion, `GraphSAGE` encoder-decoder | | |
| Edge prediction | **0.04654** | 0.04654 |
| Edge rating | 0.58012 | **0.57059** |
| *Without* graph-level fusion, `HMGSAGE` encoder-decoder | | |
| Edge prediction | **0.00358** | 0.81482 |
| Edge rating | 0.00358 | **0.66482** |

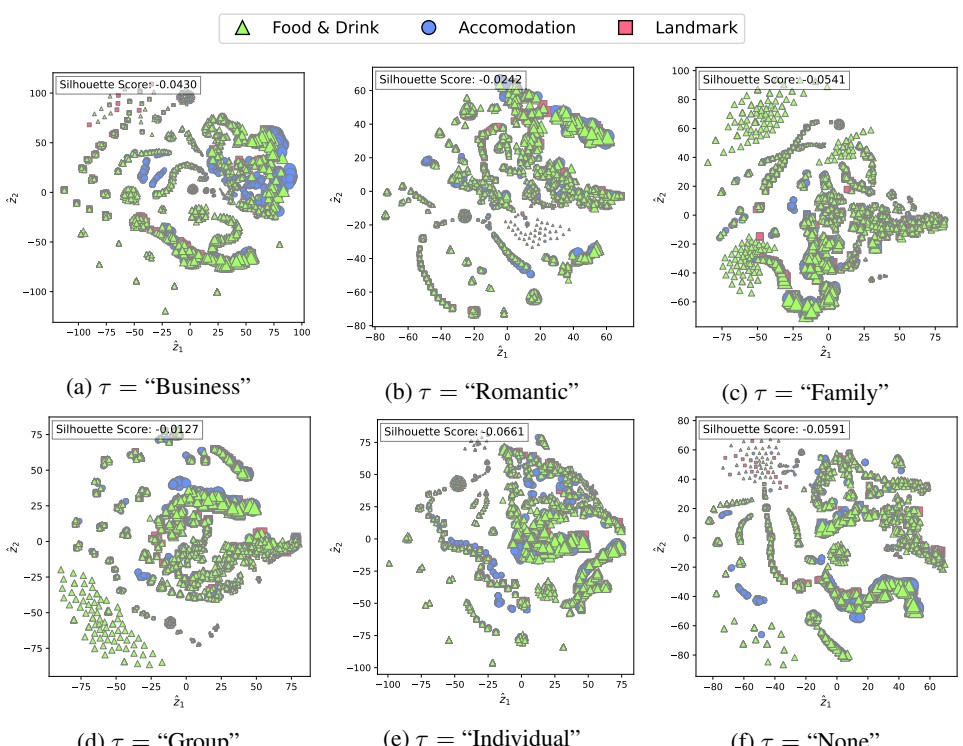

(a) $\tau = $ "Business"  (b) $\tau = $ "Romantic"  (c) $\tau = $ "Family"

(d) $\tau = $ "Group"  (e) $\tau = $ "Individual"  (f) $\tau = $ "None"

Figure 12: Two-dimensional projection of $\mathbf{z}_u$, $\forall\, \phi(u) = $ Destination for Model C3 trained without SECO. Each node's size is proportional to its degree.

with those of the `SECSAGE` model, shows a larger degree of separation between the classes when using SECO compared to Model C3 as given by the silhouette score across all multiplex layers.

Finally, we report the confusion matrices for the classification task for Model C3 trained without SECO and the `SECSAGE` model in Figure 14. We observe that the `SECSAGE` model slightly outperforms Model C3 in terms of the classification accuracy for the "Accomodation" minority class.

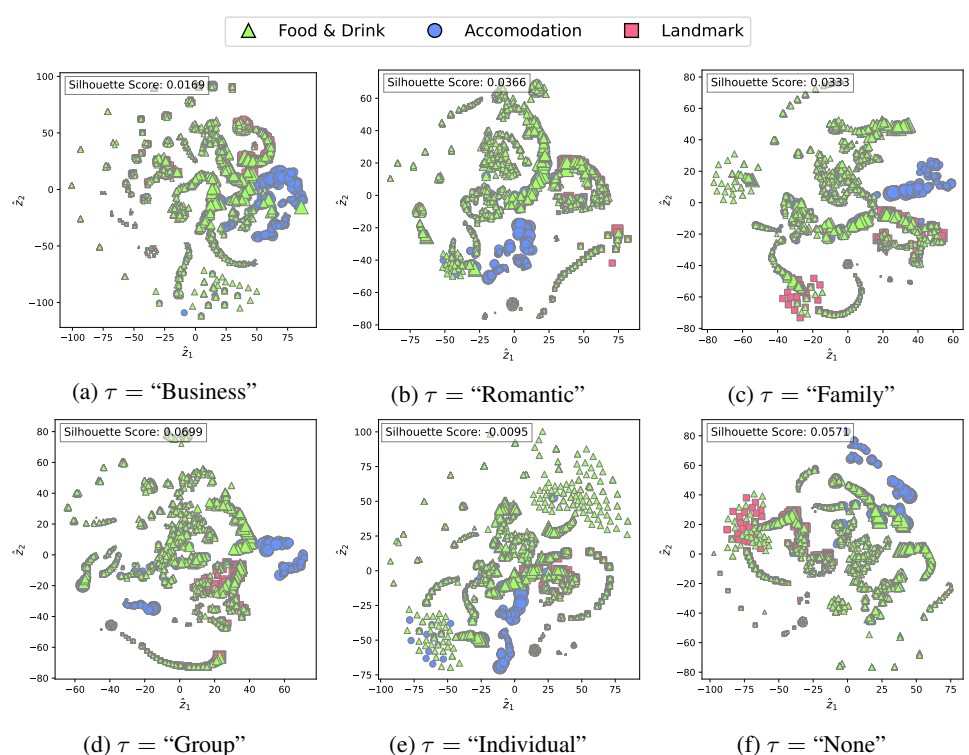

Figure 13: Two-dimensional projection of $\mathbf{z}_u$, $\forall \, \phi(u) = $ Destination for `SECSAGE` model. Each node's size is proportional to its degree.

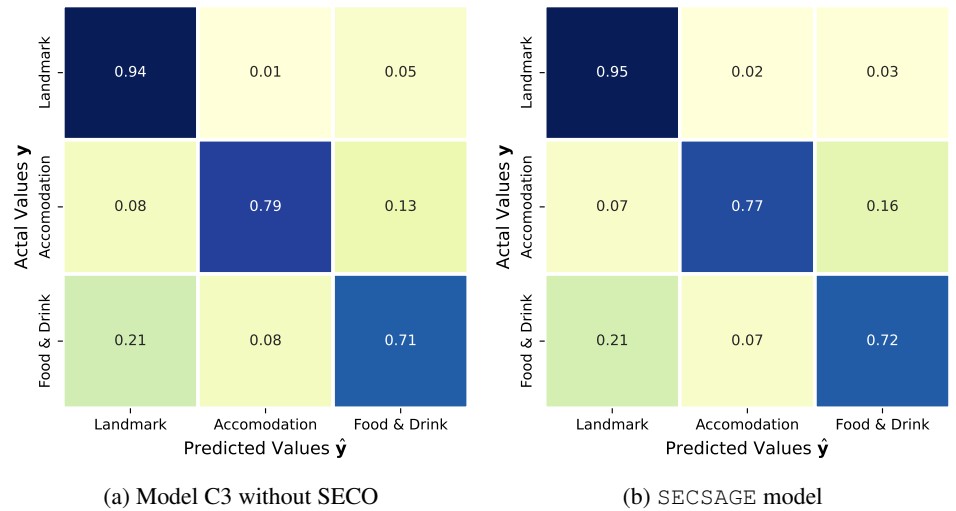

Figure 14: Confusion matrices for the classification task of the "Travel Dubai" dataset. (a) Model C3 trained without SECO and (b) `SECSAGE` trained with SECO. Both models use `soft` prediction-level fusion.

### E.2 "AMAZON" DATASET RESULTS

In this section, we report additional detailed results obtained by training the models in Section 3.2 on the "Amazon" dataset. Figures 15a and 15c show ROC and PR-RC curves, respectively for the edge prediction task. Figures 15b and 15d show the ROC and PR-RC curves, respectively for the edge rating task. We report the values of the thresholds for the `HMGSAGE` and `GraphSAGE` models

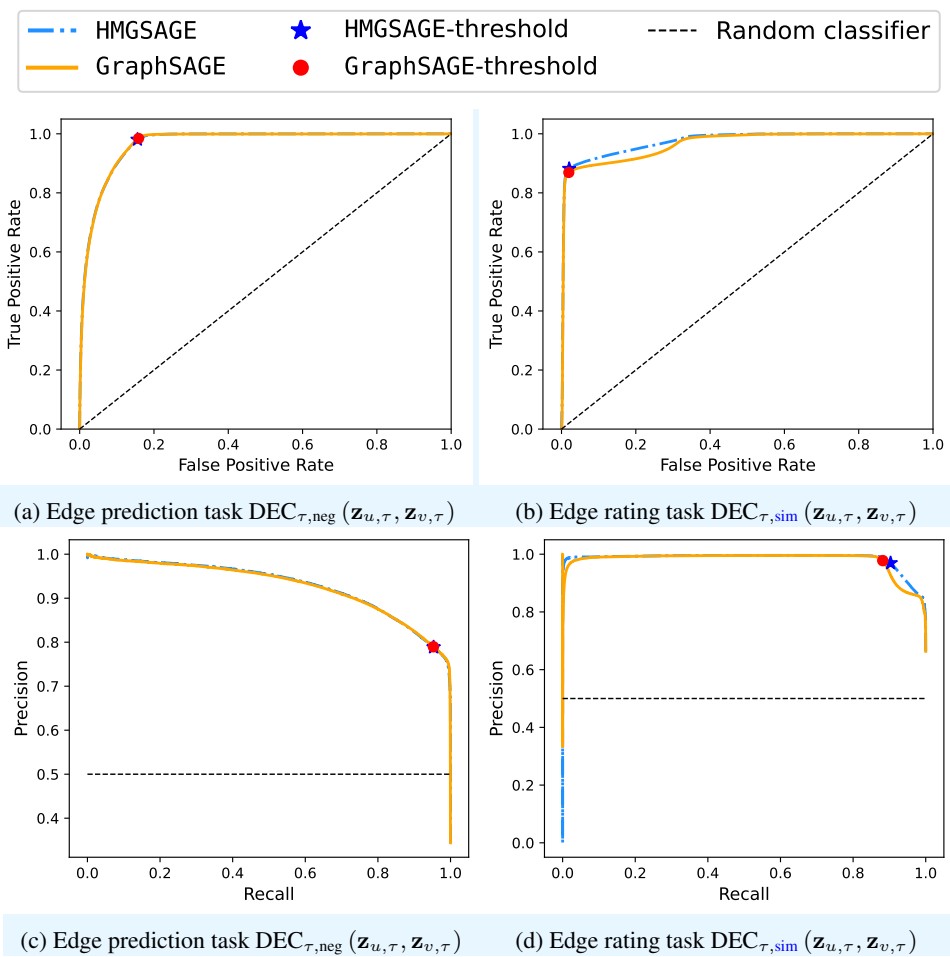

(a) Edge prediction task $\text{DEC}_{\tau,\text{neg}}\left(\mathbf{z}_{u,\tau}, \mathbf{z}_{v,\tau}\right)$    (b) Edge rating task $\text{DEC}_{\tau,\text{sim}}\left(\mathbf{z}_{u,\tau}, \mathbf{z}_{v,\tau}\right)$

(c) Edge prediction task $\text{DEC}_{\tau,\text{neg}}\left(\mathbf{z}_{u,\tau}, \mathbf{z}_{v,\tau}\right)$    (d) Edge rating task $\text{DEC}_{\tau,\text{sim}}\left(\mathbf{z}_{u,\tau}, \mathbf{z}_{v,\tau}\right)$

Figure 15: Detailed evaluation of the binary output of the GAE for the edge prediction and edge rating tasks of the "Amazon" dataset using (a,b) ROC and (c,d) PR-RC curves.

in Table 11. The confusion matrices for the classification task for Model C3 trained without SECO

Table 11: Threshold values for the GAE models trained on the "Amazon" dataset. The bold values indicate the selected threshold values for each task.

| Decoder task | Threshold AUC ROC | Threshold PR-RC |
|---|---|---|
| *With* graph-level fusion, `GraphSAGE` encoder-decoder | | |
| Edge prediction | **0.02272** | 0.02714 |
| Edge rating | 0.80440 | **0.78718** |
| *Without* graph-level fusion, `HMGSAGE` encoder-decoder | | |
| Edge prediction | **0.02311** | 0.02689 |
| Edge rating | 0.81242 | **0.78496** |

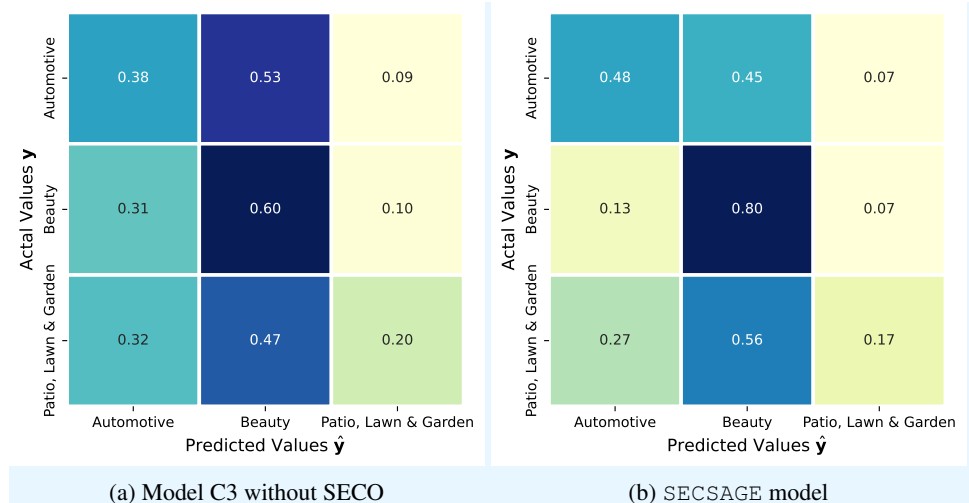

(a) Model C3 without SECO        (b) `SECSAGE` model

Figure 16: Confusion matrices for the classification task of the "Amazon" dataset. (a) Model C3 trained without SECO and (b) `SECSAGE` trained with SECO. Both models use `soft` prediction-level fusion.

and the `SECSAGE` model are shown in Figure 16. We observe that the `SECSAGE` model greatly outperforms Model C3 in terms of the classification accuracy for the "Beauty" minority class and the "Automotive" majority class.

### E.3 EFFECT OF EDGE WEIGHTED LOSS ON GAE AND CLASSIFICATION MODELS

In Sections 2.2 and 2.3, we explained that the total loss for several multiplex layer $\tau \in \mathcal{R}$ is aggregated by a weighting factor that is defined as

$$w_\tau = \frac{\sum_{\tau \in \mathcal{R}} |\mathcal{E}_\tau|}{|\mathcal{E}_\tau|}. \tag{10}$$

This means that more emphasis is placed on underrepresented multiplex layers during training reducing the possibility of overfitting on majority layers. We demonstrate the merits of this approach by running an experiment on the "Travel Dubai" dataset using the same workflow as in Section 3.3 without the weights in Equation 10, i.e., $w_\tau = 1 \; \forall \; \tau \in \mathcal{R}$ placing equal weighting on each layer. The results of these experiments on the "Travel Dubai" dataset are shown in Table 12 below. The weighted loss results reported in Table 3 are shown again in Table 12 to facilitate the comparison.

Table 12: Node classification results comparing models C3, and SECSAGE when trained using weighted and unweighted loss on the "Travel Dubai" dataset.

| Prediction-fusion | soft | hard | soft | hard |
|---|---|---|---|---|
| Graph-fusion | ✗ | ✗ | ✗ | ✗ |
| with SECO | ✗ | ✗ | ✓ | ✓ |
| Model | C3 | C3 | SECSAGE | SECSAGE |
| Weighted loss | | | | |
| Accuracy | 0.74157 | 0.74566 | 0.74566 | **0.75383** |
| Micro F1 | 0.76521 | 0.76855 | 0.76886 | **0.77615** |
| **Macro F1** | 0.69267 | 0.69407 | 0.69648 | **0.70127** |
| Unweighted loss | | | | |
| Accuracy | 0.62717 | 0.44433 | 0.74157 | 0.74872 |
| Micro F1 | 0.59656 | 0.49181 | 0.76466 | 0.76984 |
| **Macro F1** | 0.33931 | 0.33581 | 0.69288 | 0.68851 |

Similarly, the experiments are run on the "Amazon" dataset where there is a bigger imbalance between the multiplex layers "Verified purchase" and "Unverified purchase". The results are reported in Table 13 and compared against the weighted results presented in Section 3.3.

Table 13: Node classification results comparing models C3, and SECSAGE when trained using weighted and unweighted loss on the "Amazon" dataset.

| Prediction-fusion | soft | hard | soft | hard |
|---|---|---|---|---|
| Graph-fusion | ✗ | ✗ | ✗ | ✗ |
| with SECO | ✗ | ✗ | ✓ | ✓ |
| Model | C3 | C3 | SECSAGE | SECSAGE |
| Weighted loss | | | | |
| Accuracy | 0.36424 | 0.28898 | 0.44405 | **0.44516** |
| Micro F1 | 0.41348 | 0.32382 | **0.48638** | 0.48066 |
| **Macro F1** | 0.32071 | 0.26037 | **0.37059** | 0.35510 |
| Unweighted loss | | | | |
| Accuracy | 0.28826 | 0.30549 | 0.20567 | 0.22994 |
| Micro F1 | 0.32661 | 0.34537 | 0.20303 | 0.24192 |
| **Macro F1** | 0.27840 | 0.27333 | 0.20860 | 0.20994 |