# OpenReview forum: "Jointly Training Task-Specific Encoders and Downstream Models on Heterogeneous Multiplex Graphs"
_ICLR.cc/2025/Conference — Submitted to ICLR 2025_

### Official Review · Reviewer_yCoC · 2024-10-30

**Soundness:** 2
**Presentation:** 3
**Contribution:** 2
**Rating:** 5
**Confidence:** 4

**Summary:**

This paper proposes an** improved** GraphSAGE model that optimizes layer-wise embeddings during encoding while training downstream models. Experiments on a heterogeneous multiplex graph from a travel dataset show that this method outperforms models using only graph-level or prediction-level fusion, particularly for node classification tasks.

**Strengths:**

1. This paper introduces a new method with three graph learning models and a dataset. The models improve performance by maintaining the graph structure early and targeting specific tasks. This leads to better results in link prediction and node classification on complex graphs.

2. The paper has a clear structure. Each section is well-organized, and the logic flows well. Readers can easily follow the research process.

**Weaknesses:**

1. The proposed method lacks innovation as it combines existing models without introducing significant new ideas.

2. The encoder employs parallel GraphSAGE layers to sample and aggregate neighbors of nodes, but the paper does not assess the efficiency of this approach, necessitating further experiments.

3. The experiments only compare variations of the proposed model, lacking convincing evidence from comparisons with a broader range of baseline models.

**Questions:**

Can the authors provide more detailed information about the computational efficiency of the HMGSAGE encoder? Specifically, how does the use of parallel GraphSAGE layers impact the model’s overall runtime and memory usage?

---

> ### Author Response · Authors · 2024-11-28
> **Major revisions to the paper**
>
> Thank you for the feedback.
>
> 1. **Response to Weakness 1**: We have modified section 1.2 (after revisions) to more clearly address the motivations and aims of the paper. Joint training with SECO is introduced in our paper, and though several other studies have explored the idea of joint optimization, none have integrated encoders with classifiers for graph-based systems. We have also tested our data on the [‘23 Amazon Reviews dataset](https://huggingface.co/datasets/McAuley-Lab/Amazon-Reviews-2023) to show its potential. We also expanded the original presented dataset (doubling its size in terms of nodes and edges).
> 2. **Response to Weakness 2**: As we were testing on HMGs only, we followed the convention of [Bielak (2024)](https://arxiv.org/abs/2402.17906). Testing GNN-level fusion models including DMGI, HDGI, and S2MGRL could certainly improve the depth of our analysis, but in this paper we primarily look at the benefits of the incorporation of SECO rather than the choice of backbone itself. Further research could involve looking into combining any of the aforementioned models with SECO by interchanging them with the encoder used in this paper.
> 3. **Response to Weakness 3/Question 1**:Indeed, more experiments are needed to quantify the training performance of the proposed encoder and framework. In Appendix D, we now report the wall clock time for the same number of epochs for each of the proposed models in the paper. Generally, SECO and having parallel GraphSAGE layers both lead to an increase in computational time. However, the increase in computational time resulting from the addition of a layer is less than the expected increase, calculated based on a fixed computational cost per layer, suggesting diminishing marginal costs for additional layers.
> We also note that the total time needed to train an encoder+classifier is less than the time needed to train an encoder separately using contrastive approaches + time needed to train the classifier. This is because of the large cost associated with negative sampling.
> We also report their memory cost in terms of their number of parameters, which scales linearly with the number of multiplex layers $\tau\in\mathcal{R}$.

---

### Official Review · Reviewer_AE1X · 2024-11-03

**Soundness:** 2
**Presentation:** 2
**Contribution:** 2
**Rating:** 3
**Confidence:** 4

**Summary:**

This paper introduces SECSAGE, an extension of GraphSAGE for heterogeneous multiplex graphs, which refines layer-wise embeddings concurrently with downstream model training. Specifically, the model is optimized through joint encoder-classifier training, showing improvements over selected baselines on the Travel Dubai dataset. While technically robust and effective, it would be beneficial to include experiments on other datasets, code for reproducibility, expanded baseline comparisons, and improved writing clarity.

**Strengths:**

1. The topic is interesting and relevant, addressing challenges in representation learning for heterogeneous multiplex graphs.
2. SECO combines prediction-level fusion with simultaneous optimization, resulting in improved task alignment in embeddings.
3. The proposed model demonstrates improvement over selected baselines.
4. The model’s performance is validated with multiple metrics, including accuracy, F1, and ROC-AUC.

**Weaknesses:**

1. The reliance on GraphSAGE extensions and standard fusion techniques limits the novelty of the proposed architecture.
2. The absence of code hinders reproducibility, reducing accessibility for researchers who may want to apply the proposed approach.
3. Evaluating on only a single dataset may not capture sufficient diversity, thus limiting generalizability.
4. While the model outperforms the chosen baselines, authors should expand the comparison to include a wider range of recent and diverse approaches.
5. The manuscript would benefit from further refinement to improve clarity and readability.

**Questions:**

1. Are there specific models you believe should be included in the baseline comparisons, especially in the context of heterogeneous/heterogeneous multiplex/knowledge graphs?
2. Do you have plans to make the code available publicly?
3. Since the model is evaluated on a single dataset, can you discuss the potential challenges in applying your model to other datasets?
4. Could you further clarify the novelty of the added components and explain how these modifications specifically address the unique challenges of heterogeneous multiplex graphs?

---

> ### Author Response · Authors · 2024-11-27
> **Major revisions to the paper**
>
> Thank you for your feedback and comments. We would like to draw your attention to a possible mixup in the review process. We think that the above review should belong to [submission 10400](https://openreview.net/forum?id=kp8T7G9hIh), and that [this review](https://openreview.net/forum?id=kp8T7G9hIh&noteId=06ee8Ke7pu) by reviewer vLoA is intended for us.
>
> As a result, we respond to reviewer [vLoA](https://openreview.net/forum?id=kp8T7G9hIh&noteId=06ee8Ke7pu) below:
>
> 1. **Response to Weakness 1/ Question 4**: We aimed to address this further in section 1.2 by further emphasizing the gap regarding the study of SECO, and the increased separability of our embeddings.  We work primarily with GraphSAGE due to its ability to effectively and expressively encode large input graphs, and to work inductively (unlike GNN, GCN, etc.). However, the SECO paradigm that we introduce is compatible with any GNN-adjacent backbone.
> SECO is applicable to more than just HMGs, and our comparison in Table 3 aims to address this by comparing models using graph-level fusion. We find that the benefits of SECO are even more pronounced on flattened graphs. In the context of HMGs, we find that it especially improves the quality of the embeddings, which often fail to meaningfully aggregate information across layers. Optimizing in conjunction with a classifier allows the model to decide exactly what information is useful in the encoding process, as evidenced by the difference in silhouette scores of the labeled embeddings. We expanded section 1.2 to address this information.
> Additional computational results on the Amazon product review dataset suggest that the improvement in embedding quality due to SECO is even more pronounced on datasets with an imbalanced number of edges per multiplex layer as shown in Table 3.
> The details of the effect of this imbalance are discussed in *Response 4* below.
> We hope that the lessons learned from this work can prove beneficial to the community at large from an application standpoint when working with recommendation systems, and is our reason for sharing this work with the community.
>
> 2. **Response to Question 1**: As mentioned, it would be relevant to compare SECSAGE to more HMG-specific encoder models, such as DMGI, HDGI, MHGCN, etc. Further research could both compare SECSAGE to architectures using these backbones, and also measure the effectiveness of integrating them with SECO. We aimed to use a similar approach to [Bielak (2024)](https://arxiv.org/abs/2402.17906) in our testing.
> We also tested our models on the Amazon Reviews dataset to better display their generalizability and effectiveness.
>
> 3. **Response to Weakness 2/Question 2**: Absolutely. We have been organizing our repository after the large number of experiments conducted, and will post a link to the anonymized GitHub repository in the comments section soon. We also plan to share our Travel dataset (which we have doubled in size for this revision of the paper) on Hugging Face after the review process has concluded to allow the network and graph community to use it for benchmarking and further research. We also hope to continue collecting data and expand it further.
> We will modify the camera-ready version of the paper with links to these items and share the code and data publicly whether the paper is accepted or not.
>
> 4. **Response to Weakness 3/Question 3**: As mentioned earlier, we have applied the framework to the [Amazon product review dataset](https://huggingface.co/datasets/McAuley-Lab/Amazon-Reviews-2023). The main challenge with such a dataset was its larger size (going from a graph with $\approx$ 20K nodes and edges to $\approx$ 1.5M nodes and edges).
> Training was scalable and manageable on a workstation GPU and CPU.
> However, and as mentioned in response to your earlier questions, multiplex layer imbalance can be a challenge (seen in the Amazon dataset). We applied a weighting factor to the loss terms associated with each edge, which significantly improved the training performance of the model.
> A new Appendix section E3 has been added to the paper to discuss this topic.
>
> 5. **Response to weakness 5**: As explained to reviewer [uuQc](https://openreview.net/forum?id=o0X0CPl320&noteId=MKgoQOdYc1), we have heavily edited and reorganized various sections of the manuscript to improve readability and structure.

---

### Official Review · Reviewer_uuQc · 2024-11-07

**Soundness:** 2
**Presentation:** 2
**Contribution:** 2
**Rating:** 3
**Confidence:** 5

**Summary:**

This paper proposes a novel method for jointly training task-specific encoders and downstream models on heterogeneous multiple graphs (HMG). The author first defines key concepts such as heterogeneous multiple graphs, inductive learning, and simultaneous encoder-classifier optimization SECO. Subsequently, they designed an encoder called HMGSAGE, which can handle graph data with multiple node and edge types. Building on this foundation, the author further expanded HMGSAGE and constructed a graph autoencoder GAE capable of inductive link prediction and graph structure data node classification. The core contribution of this paper is the introduction of the SECO paradigm, which learns the optimal node embeddings for the downstream task through supervised learning by iteratively and simultaneously updating the encoder and downstream classifier, thereby improving the performance of the downstream task.
advantage:

**Strengths:**

1.This paper proposes a new joint training framework that closely integrates the training process of the encoder and downstream models, providing an effective solution for tasks on heterogeneous multiple graphs.

2.The author clearly defines and explains concepts such as heterogeneous multiple graphs, inductive learning, and SECO, providing a solid theoretical foundation for readers.

**Weaknesses:**

1.The writing quality of the entire paper is relatively lacking, with problems such as unclear expression and confusing logic, especially in the introduction section, where there are significant writing issues. The discussion of the recommendation system in the introduction is too long and not effectively connected to the core research content and motivation of the paper, making it difficult for readers to understand the actual significance and motivation of the research.

2. The overall organizational structure of the paper is rather confusing, leading to confusion for readers in understanding. There is a lack of clear logical connection between the various parts, making it difficult to trace.

3. The method proposed in this article is essentially a stack of existing technologies. Specifically, the author uses GraphSAGE as an encoder, stacks it with a joint training method based on SECO, and introduces the GAE model structure for learning. Although these technologies are effective in themselves, they do not demonstrate sufficient innovation and lack new theoretical or practical value.

4. Only one dataset was used for verification in the experimental part, and there was no sufficient comparative experiment. In particular, other methods listed in Table 1 mentioned in the paper were not effectively compared as a baseline for comparison.

**Questions:**

Please refer to the weaknesses.

---

> ### Author Response · Authors · 2024-11-27
> **Major revisions to the paper**
>
> Thank you for your feedback on our paper, which has helped us significantly address its weaknesses. We respond to each identified weakness below:
>
> 1. We understand that the introduction and related works sections were long and unclear. Each has been cut down significantly to reduce the paper’s length and improve its conciseness while maintaining the flow of the paper. We also add revisions to more clearly address the motivations and significance of the research in section 1.2. We discuss how these motivations were addressed in the paper in the conclusion section.
>
> 2. We have significantly edited and restructured each section to make the paper flow better. We have moved some definitions and notations to the methodology section where they are used.
>
> 3. As mentioned above, we have modified section 1.2 (after revisions) to more clearly address the motivations and aims of the paper. Joint training with SECO is introduced in our paper, and though several other studies have explored the idea of joint optimization, none have integrated encoders with classifiers for graph-based models so we share this result with the GRL community. We have also tested our models on the ‘23 Amazon Reviews dataset to show its generalizability.
>
> 4. We have performed additional experiments using another benchmark dataset based on Amazon product review data, which is significantly larger than the initially presented dataset. We also expanded the original presented dataset (doubling its size in terms of nodes and edges). With regard to the baselines, we have clarified in our response to other reviewers that it is indeed lacking. However, our objective is to understand the effect of simultaneously training the encoder and downstream models, rather than the effect of the encoder itself on the quality of the learned embeddings. Our encoder could be easily interchanged with the others presented in Table 1, and it is certainly a potential topic for future research.

---

### Author Response · Authors · 2024-11-27
**Major revisions to the paper**

We thank the reviewers for their thorough comments which have helped us improve the manuscript. We respond to the reviewers’ comments below. We have revised the manuscript by correcting a few minor errors and making adjustments where necessary to improve its flow and readability. We also added new experiments to demonstrate the framework's generalizability. All changes to the manuscript are highlighted through the use of $\color{blue}\textbf{blue}$ fonts to help facilitate the review.

---

> ### Author Response · Authors · 2024-11-29
> **Summary of main revisions to the paper**
>
> We would like to provide an overview of the main revisions to assist the reviewers and ACs with the review process.
>
> * We heavily edited the introduction to make it more concise and convey the message of the paper clearly. We have also dedicated a part of it to clearly outline the contributions of the paper in Section 1.2 which are:
>
>     * To apply joint-optimization techniques for learning task-specific representations for graph-based recommendation systems.
>     * Handle the challenges of learning meaningful representations in the context of Heterogeneous Multiplex Graphs (HMGs).
>     * Provide a set of best practices for learning representations from realistic graph datasets.
>
> * Conducted new experiments on an expanded version of the presented dataset. We have doubled the number of nodes and edges and collected new data, which we hope to publish and share publicly as well.
> * Conducted new experiments on a benchmark dataset for Amazon product reviews found [here](https://huggingface.co/datasets/McAuley-Lab/Amazon-Reviews-2023). Thanks to the reviewers' feedback and comments, these experiments have greatly helped us test our framework on large datasets, shows its scalability, and gather some new interesting insights when working with imbalanced HMGs, where one multiplex layer is underrepresented. Please continue to the bottom of this comment to read more about that.
>
> ## How we addressed each of the above contributions:
> * **Contribution 1:** We demonstrate how joint optimization of embeddings for downstream tasks improved performance on said tasks using controlled experiments and various datasets. We term the application in this paper: Simaltaneous Encoder Classifier Optimization (SECO). While intuitive, we felt that there is no systematic study (especially in the context of graph-based systems) to quantify this effect and interpret it in the embedding space. We show in the paper that this consistently results in a performance improvement on downstream tasks and can be explained in the embedding space by examining the silhouette scores of the embedded data (which shows an improvement all across the board ($\mathbf{z}_\tau \forall \tau\in\mathcal{R}$) in Figures 4 and 13. This could be a step towards more explainable embeddings and AI.
> * **Contribution 2:** We demonstrate that confining the representation learning process to specific multiplex layers $\tau \in \mathcal{R}$ and learning seperate embeddings $\mathbf{z}_\tau\forall\tau\in\mathcal{R}$ helps improve the expressiveness of said embeddings and their combination/aggregation in downstream tasks. Again we demonstrate this through the use of systematic experiments on both a GAE and a classification task. This is what led us to develop $\texttt{HMGSAGE}$ and $\texttt{SECSAGE}$
> * **Contribution 3:** Thanks to the reviewers' suggestion to test the framework on other larger datasets, we have discovered new insights which we would like to share. In datasets with imbalanced multiplex layers, i.e., $\lvert \mathcal{E}\_{\tau_1} \rvert \neq \lvert \mathcal{E}\_{\tau} \rvert$, where $\tau_1,\tau_2 \in \mathcal{R}$, we noticed poor test performance on downstream tasks. We discovered that normalizing the loss terms by the layer size (in terms of the number of edges $\lvert\\mathcal{E}_{\tau}\rvert$ greatly improved test performance and led to better generalizability. We dedicated appendix section E2 to discuss this and referred to it in the paper. Here is a summary of this result from the paper with the F1 score increasing from 0.20860 to 0.37059. Training on the flattened graph also yielded poor performance as seen in Table 3 (hovering around 0.2). This means that our approach of compartmentalizing the learning process could have some practical benefits.
>
> | **Prediction-fusion** | `soft`       | `hard`       | `soft`       | `hard`       |
> |------------------------|--------------|--------------|--------------|--------------|
> | **Graph-fusion**       | ❌           | ❌           | ❌           | ❌           |
> | with SECO             | ❌           | ❌           | ✅           | ✅           |
> |**Weighted Loss**||||
> | **Model**      | **C3**      | **C3**      | **$\texttt{SECSAGE}$**  | **$\texttt{SECSAGE}$**  |
> | Accuracy       | 0.36424     | 0.28898     | 0.44405      | **0.44516**  |
> | Micro F1       | 0.41348     | 0.32382     | **0.48638**  | 0.48066      |
> | **Macro F1**   | 0.32071     | 0.26037     | **0.37059**  | 0.35510      |
> |**Unweighted Loss**||||
> | **Model**      | **C3**      | **C3**      | **$\texttt{SECSAGE}$**  | **$\texttt{SECSAGE}$**  |
> | Accuracy       | 0.28826     | 0.30549     | 0.20567      | 0.22994      |
> | Micro F1       | 0.32661     | 0.34537     | 0.20303      | 0.24192      |
> | **Macro F1**   | 0.27840     | 0.27333     | 0.20860      | 0.20994      |
>
> ### Closing remark:
> Overall this has been a great learning experience for us, regardless of the outcome and we wish to thank the reviewers for their dedication and very useful feedback.

---

### Meta-Review · Area_Chair_AbUP · 2024-12-10

**Metareview:**

This paper extends GraphSAGE for representation learning on Heterogeneous Multiplex Graphs by refining layer-wise representations. Although it is an interesting topic, the novelty and contributions are very limited by leveraging existing GraphSAGE and fusing techniques. Besides, the evaluations are performed only on one dataset, and the baselines are old. Thus, it is not fully justified. Although authors provide the improvements on the expression, it also much more efforts to enhance the quality and clarity.

**Additional Comments On Reviewer Discussion:**

Since the feedback is posted at the end of the rebuttal, there is no discussion between authors and reviewers. By checking the author's feedback, I think most of the concerns raised by the reviewers, especially reviewer uuQc who provides negative ratings, are not alleviated.

---

### Decision · Program_Chairs · 2025-01-22

Reject